# A toolbox for black hole scattering

**Nava Gaddam and Nico Groenenboom**

*Institute for Theoretical Physics and Center for Extreme Matter and Emergent Phenomena, Utrecht University, 3508 TD Utrecht, The Netherlands.*

*E-mail:* gaddam@uu.nl, n.groenenboom@uu.nl

ABSTRACT: Hawking's free field theory is expected to break down after Page time. In previous work, we have shown that a primary dynamical reason for this breakdown is the dominance of graviton fluctuations of the horizon that mediate scattering processes. In this article, we present a toolbox for such 'black hole scattering' computations. The toolbox comprises of explicit expressions for the graviton propagator near the horizon in an angular momentum basis for all angular momentum modes of either parity, the leading interaction rules, and most importantly a rewriting of the theory in terms of a scalar theory with an interesting four-vertex. We demonstrate how this rewriting drastically reduces the number of diagrams to be calculated in the original formulation. Finally and perhaps most remarkably, we observe that the black hole entropy appears to emerge from the multiplicity of external legs of the dominant $2 \to 2N$ amplitudes in this theory.

# 1  Introduction

Hawking argued that black holes violate predictability in the laws of physics when free scalar fields propagate on a black hole background [1, 2]. A natural question to ask is whether this conclusion would change if the scalar fields were to interact with the inevitably (quantum) fluctuating black hole spacetime. To answer such a question, it is imperative that calculations in the presence of a fluctuating black hole horizon are doable. We have shown in recent years that several relevant calculations can be done [3, 4, 5] that teach us interesting (near-horizon) physics. It is the aim of the present article to summarise these developments, extend them, and present a comprehensive toolbox for further calculations.

The calculations of interest are those of scattering amplitudes. The predominant question of interest is, can we calculate the amplitude of $N$ particles emerging from a Schwarzschild black hole when $M$ particles are thrown into it? The interactions governing the scattering processes are mediated by graviton fluctuations of the black hole horizon. To leading order, these gravitons couple to the stress tensor of the minimally coupled scalar field resulting in three-vertices of interest. Extension to higher order interactions is both desirable and straightforward with the tools we develop, although explicit calculations are likely to be hard to do in complete generality.

Several questions may arise when scattering amplitudes in curved spacetimes are considered. We list some of them that are rather natural in the context of black holes here and provide conceptual answers:

- Black holes break translational invariance. How could we then possibly compute Feynman diagrams in momentum space? While four-momenta are indeed hard to define, exploiting the spherical symmetry of the background, they can be traded for two-momenta (along the longitudinal directions) and angular momentum labels (after integrating the sphere out). Of course, the longitudinal part of the metric is not flat, but its conformal flatness allows us to trade the curvature for potentials. And the longitudinal momenta are defined after this Weyl rescaling, without loss of generality.

- How are kinematic variables defined if the spacetime is curved? After the Weyl rescaling described above to exploit the conformal flatness of the longitudinal part of the Schwarzschild metric, kinematic variables are defined with the help of the longitudinal two-momenta arising from the flat part of the metric. The price we pay is of course that the curvature effects have to be traded for potentials.

- The black hole contains a large number of degrees of freedom. But perturbation theory is likely to breakdown when scattering processes involving many external legs are considered. So, are we likely to learn much at all from black hole scattering? While perturbation theory with many external legs does breakdown in the vacuum [6], in the presence of a horizon, the size of the black hole results in an emergent coupling constant $\gamma := \kappa/R = \sqrt{8\pi G}/R$, where $R$ is the Schwarzschild radius [7, 3, 4]. For large black

holes, the range of validity of perturbation theory in the black hole background is much wider [3, 4], and considering many external particles in fact results in the emergence of important time scales (like Page time) [5].

- Isn't scattering on a black hole background governed by the Dray-'t Hooft shockwave? Just as in its flat space counterpart [8, 9], the on-shell part of the elastic $2 \to 2$ amplitude contains the Dray-'t Hooft shockwave [10, 11, 12, 13, 14, 15]. However, unlike in the case of flat space, there are off-shell graviton perturbations that do not satisfy the classical equations of motion but do contribute to the eikonal amplitudes [4]. Perhaps more interestingly, particle production in amplitudes has no known counterpart as a classical solution to Einstein's equations. Therefore it is fair to say that these scattering amplitudes capture far more information than is contained in the shockwave.

- Isn't gravity (perturbatively) non-renormalisable? How far can computing amplitudes take us? While it is true that ultraviolet physics will kick in at some stage, one of the remarkable features of black hole scattering is that as long as centre of mass energies of scattering satisfy $E \gg \gamma M_{Pl}$, ultraviolet effects are heavily suppressed. For large black holes, $\gamma$ is very small and this relation has a wide range of validity. In fact, we expect the scattering to be of high energy near the horizon due to the large redshift. Therefore, there is a lot more to be understood in the infrared than one might have expected. These are all effects that can be explicitly calculated as deviations from Hawking's free field theory.

With those questions answered, we now list a few results of black hole scattering that may be considered as significant successes.

**Successes of black hole scattering to date**

- **Black hole eikonal:** As alluded to above, the emergence of a new eikonal phase near the horizon that is different from, but somewhat analogous to, the flat space eikonal is noteworthy. In flat space, the eikonal phases emerges at large impact parameters when scattering energies are trans-Planckian $E \gg M_{Pl}$. In the black hole eikonal, however, the range of validity is much wider owing to a suppression by the size of the black hole: $E \gg \gamma M_{Pl}$.

- **Black hole scattering > shockwave:** The Aichelburg-Sexl shockwave [16] in flat space is equivalent to the flat space eikonal ladder which contains no off-shell fluctuations [17]. The black hole eikonal ladder, on the other hand, contains the Dray-'t Hooft shockwave [18] but also contains off-shell virtual graviton fluctuations that carry more information than the on-shell classical solution [4]. While the significance of this difference is not entirely understood, it is certainly evident that black hole scattering carries more information than the shockwave. Moreover, inelastic processes in black hole scattering [5] have no known shockwave counterparts.

- **Emergence of time scales:** Black holes have interesting time scales associated with them, scrambling time and Page time are universal examples. One of the significant successes of the black hole scattering program is the natural emergence of these time scales from explicit calculations of amplitudes. Given the scattering matrix associated with a certain process, there is a canonical time scale (the Eisenbud-Wigner time delay [19, 20]) that estimates the time that ingoing particles 'spent' in the scattering region. For elastic $2 \to 2$ black hole scattering, this time delay turns out to be scrambling time [5]. Particle production on the other hand, via a computation of $2 \to 2N$ scattering, requires Page time [5]. This statement in particular implies that Hawking's free field theory on a fixed background is explicitly invalid after Page time because interactions mediate information return.

- **Emergence of black hole entropy:** In a scattering matrix approach to quantum black hole physics, it has not been clear how a coarse-grained entropy may emerge. In the present article, we observe that the black hole entropy appears to naturally arise from the multiplicity of the external legs of the dominant $2 \to 2N$ scattering.

- **Observational consequences:** The inspiral phase of the observed gravitational wave signals from compact binary mergers are well approximated by scattering in flat space at large impact parameters [21, 22, 23].[1] On the other hand, black hole scattering implies that modes that are ingoing near the horizon create virtual gravitons that then release outgoing modes that leak to the outside of the classical gravitation potential in the form of gravitational echoes [26]. These modify the familiar quasi-normal mode spectrum and provide a model independent prediction for echoes from black holes. Gravitational echoes were previously thought to emerge from exotic compact objects (ECOs) [27]. However, gravitational echoes from black hole scattering will reflect on the black hole memory effect. In a broader sense, it is fair to expect that black hole scattering will contribute to the gravitational wave signals of the post-merger phase of the compact binary mergers observed in nature. An estimation of the significance of this contribution would go a long way towards an understanding of observational signatures of black holes in nature. Finally, in analogy to the Post-Newtonian and Post-Minkowskian expansions associated with the inspiral phase, a natural Post-BH expansion possibly governs the post-merger phase that improves upon the classical ringdown prediction.

The success of black hole scattering is not all-encompassing. There certainly are shortcomings, some of which may be more important and/or harder to rectify than others.

**Some shortcomings so far**

- In its strict sense, it is not clear how one must properly define the S-matrix in the presence of a black hole. In particular, how Hawking radiation may be incorporated into the asymptotic out states is an interesting concern. We essentially turn a blind

---

[1]See [24, 25] for recent reviews.

eye to this issue, and compute scattering amplitudes as if asymptotic states were well-defined. A consequence of this is that while the corrections to Hawking's picture are directly calculable in this approach, incorporating Hawking's leading order Bogoliubov transformations require additional care.

- An analytic expression for the graviton propagator in a black hole background is unavailable. Therefore, we resort to an approximate version in the near-horizon region. For several observables, this sufficiently captures the dominant physics at low energies. Nevertheless, it would be of particular interest to understand the limitations of this approximation with care.

- Analyses of scattering processes have been limited to the leading three-vertex arising from the graviton coupled to the minimally coupled scalar stress-tensor. Vertex corrections, classical corrections from graviton self-interactions, higher order interactions, other gauge and matter fields have all yet to be studied.

- Another obvious shortcoming of the black hole scattering program is a technical one. All calculations so far have only been done in the presence of a large black hole. Calculations in the presence of a time dependent background are analytically intractable. While the scale at which these effects become is not entirely clear, it is certainly of interest to exhaustively understand what questions necessitate these effects to be studied.

**What is new in the present article?** In [3, 4], we have derived the graviton propagator for the even parity modes in the Regge-Wheeler gauge [28] in the near horizon region and computed elastic $2 \rightarrow 2$ scattering amplitudes in a black hole eikonal limit alluded to above. It is known that there are additional gauge redundancies in the low angular momentum modes [3, 4, 29, 30]. Fixing this additional gauge redundancy in the even parity mode of the s-wave, we have computed $2 \rightarrow 2N$ tree-level amplitudes and an infinite class of loop corrections thereof in [5]. In the present article, we arrive at the following results:

- We derive the graviton propagator in the near horizon region for all modes of the graviton (even and odd parity and all angular momentum modes with both even and odd parity). This is done in Section 2.

- Considering three-vertex interactions arising from the graviton coupling to the stress tensor of a minimally coupled scalar field, we show that certain modes of the graviton do not contribute to the physical S-matrix at this level. This is presented in Section 3.

- In Section 4, we show that the non-interacting mode can be decoupled from the theory and integrated out. Consequently, we arrive a rewriting of the theory in terms of an effective scalar theory with a four-vertex (without loss of generality) that captures $2 \rightarrow 2$ amplitudes involving the original graviton exchange.

- In Section 5, we show why the new rewriting of the theory in terms of a scalar four-vertex is particularly efficient for computations. For instance, 972 diagrams that contribute

to a three-loop $2 \to 4$ amplitude are captured by all of only four topologically distinct diagrams in the rewritten theory.

- Finally, in Section 6, we observe that the entropy of the black hole naturally seems to arise from the multiplicity of the dominant $2 \to 2N$ scattering.

## 2 Graviton perturbations in partial waves

The quadratic action for graviton perturbations (say $h_{\mu\nu}$) about a spherically symmetric vacuum solution (say $g_{\mu\nu}$) to Einstein's equations, with $\bar{g}_{\mu\nu} = g_{\mu\nu} + \kappa h_{\mu\nu}$ and $h = g^{\mu\nu} h_{\mu\nu}$, can be written as [31, 4]

$$
\begin{aligned}
S &= -\frac{1}{2} \int \mathrm{d}^4 x \sqrt{-g}\, h^{\mu\nu} \left[ \frac{1}{2} \left( 2\nabla^\rho \nabla_{(\mu} h_{\nu)\rho} - \Box h_{\mu\nu} - \nabla_\mu \nabla_\nu h \right) - \frac{1}{2} g_{\mu\nu} \left( \nabla^\rho \nabla^\sigma h_{\rho\sigma} - \Box h \right) \right] \\
&= -\frac{1}{4} \int \mathrm{d}^4 x \sqrt{-g} \left( h^{\mu\nu} - \frac{1}{2} g^{\mu\nu} h \right) \left( 2\nabla^\rho \nabla_{(\mu} h_{\nu)\rho} - \Box h_{\mu\nu} - \nabla_\mu \nabla_\nu h \right) + \frac{\kappa}{2} \int \mathrm{d}^4 x\, h^{\mu\nu} T_{\mu\nu},
\end{aligned}
\tag{2.1}
$$

where we wrote the action in terms of the first order variation of the Einstein tensor (square parenthesis in the first line) and in terms of the first order variation of the Ricci tensor (the second parentheses in the second equality) in the second line. The background metric of interest is

$$
\mathrm{d}s^2 = -2A(r)\mathrm{d}x\mathrm{d}y + r^2(x,y)\,\mathrm{d}\Omega_2^2 \quad \text{with} \quad A(r) = \frac{R}{r} e^{1-\frac{r}{R}}. \tag{2.2}
$$

We discuss the choice of coordinates, and the non-vanishing Riemann tensor components for this Schwarzschild metric in Appendix A. To exploit the spherical symmetry of the background, we now expand the graviton in spherical harmonics as in [28]:

$$
h_{\mu\nu} = \sum_{\ell,m} h^-_{\ell m,\mu\nu} + \sum_{\ell,m} h^+_{\ell m,\mu\nu}, \tag{2.3}
$$

where $h^-_{\ell m,\mu\nu}$ are the so-called odd graviton modes and $h^+_{\ell m,\mu\nu}$ the even modes. It can be checked that these even and odd modes decouple in the quadratic action allowing us to consider them separately [28, 4].

### 2.0.1 The odd harmonics

The odd harmonics are parametrised as

$$
h^-_{\ell m,\mu\nu} = \begin{pmatrix} 0 & 0 & -h^-_x \csc\theta \partial_\phi & h^-_x \sin\theta \partial_\theta \\ & 0 & -h^-_y \csc\theta \partial_\phi & h^-_y \sin\theta \partial_\theta \\ & & h_\Omega \csc\theta \left( \partial_\theta \partial_\phi - \cot\theta \partial_\phi \right) & \frac{1}{2} h_\Omega \left( \csc\theta \partial_\phi^2 + \cos\theta \partial_\theta - \sin\theta \partial_\theta^2 \right) \\ & & & -h_\Omega \sin\theta \left( \partial_\theta \partial_\phi - \cot\theta \partial_\phi \right) \end{pmatrix} Y_\ell^m. \tag{2.4}
$$

Using index notation, this can be written as

$$
\begin{aligned}
h_{ab}^- &= 0 \\
h_{aA}^- = h_{Aa}^- &= h_a \eta_A \\
h_{AB}^- &= h_\Omega \hat{\nabla}_{(A} \eta_{B)} \,,
\end{aligned}
\tag{2.5}
$$

where the indices $(a, b)$ run over the longitudinal coordinates and $(A, B)$ take values in the transverse spherical directions. Of course, as this is a partial wave expansion, all functions are only functions of the longitudinal coordinates with the angular dependence explicitly extracted out. Furthermore, we defined $\hat{\nabla}_A$ to be the covariant derivative on the unit two-sphere and pseudo-tensor

$$
\eta_A = -\epsilon_A{}^B \partial_B Y_\ell^m
\tag{2.6}
$$

on the two-sphere that is characteristic for the odd modes. As can be readily checked, $h_\Omega$ falls away for the $\ell = 0, 1$ modes and in what follows, we will remove it by gauge choice (for the $\ell > 1$ modes), leaving us with only $h_{aA}^- = h_a \eta_A$ to consider. Therefore, the quadratic action for the odd harmonics $h_a^-$ can be written as

$$
S = \frac{\lambda - 1}{2} \sum_{\ell, m} \int \mathrm{d}^2 x \, h^a \left( \eta_{ab} \left( \partial^2 - \mu^2 \left( \lambda - \frac{3}{2} \right) \right) - \partial_a \partial_b + 3\mu^2 x_{[a} \partial_{b]} \right) h^b \,,
\tag{2.7}
$$

with $\mu = R^{-1}$ and $\lambda = \ell^2 + \ell + 1$. Several manipulations had to be performed to arrive at this action, all detailed in Appendix C. We first plug in the odd graviton $h_{aA}^- = h_a \eta_A$ into the quadratic action (2.1), then write the covariant derivatives out before integrating over the sphere to reduce the theory to two dimensions. Then, we make a field redefinition $h_a \to \sqrt{A(x, y)} \, \mathfrak{h}_a$ to exploit the conformal flatness of the two dimensional metric $g_{ab} = A(x, y) \eta_{ab}$ which allows us to trade all covariant derivatives for partial derivatives and potentials. Furthermore, since the quadratic operator is still not easily invertible notwithstanding these manipulations, we make a near-horizon approximation where $r \sim R$ implying $x, y \ll R$ and $A(x, y) \sim 1$. This yields the above action. As we will show in Section 2.2, this action is reliable only for the multipole modes $\ell \geq 2$. For the dipole $\ell = 1$, there is additional gauge redundancy that needs care.

Finally, it is worth noting that the action is proportional to $\lambda - 1$ and therefore vanishing for the $\ell = 0$ or $\lambda = 1$ mode, since there is no odd harmonic in the monopole mode. Moreover, the action grows rapidly for larger multipole moments rendering them to be subleading.

### 2.0.2 The even harmonics

The even harmonics are parametrised by

$$
h_{\ell m, \mu\nu}^+ = \begin{pmatrix} H_{xx} & H_{xy} & h_x^+ \partial_\theta & h_x^+ \partial_\phi \\ & H_{yy} & h_y^+ \partial_\theta & h_y^+ \partial_\phi \\ & & r^2(K + G\partial_\theta^2) & r^2 G(\partial_\theta \partial_\phi - \cot\theta \partial_\phi) \\ & & & r^2(K \sin^2\theta + G(\partial_\phi^2 + \sin\theta\cos\theta\partial_\theta)) \end{pmatrix} Y_\ell^m .
\tag{2.8}
$$

Using index notation this can be written as

$$h_{ab}^+ = H_{ab}Y_\ell^m \tag{2.9}$$

$$h_{aA}^+ = h_{Aa}^+ = h_a^+ \partial_A Y_\ell^m \tag{2.10}$$

$$h_{AB}^+ = Kg_{AB}Y_\ell^m + G(\partial_A\partial_B - \Gamma_{AB}^C\partial_C)Y_\ell^m. \tag{2.11}$$

The $G$ and $h_a^+$ fields do not contribute to the monopole ($\ell = 0$) and we will remove them by a choice of gauge for the dipole and multipole modes ($\ell \geq 1$). Just as in the case of the odd harmonics, this allows us to only consider $h_{ab}^+ = H_{ab}Y_\ell^m$ and $h_{AB}^+ = Kg_{AB}Y_\ell^m$ without any loss of generality. Following steps identical to what we did for the odd harmonics, first integrating out the two sphere, redefining the fields as $H_{ab} \to \mathfrak{h}_{ab}$ and $K \to \mathcal{K}$, and making the near horizon approximation, the following quadratic action for the even harmonics can be derived

$$S = \frac{1}{4} \int d^2x \left( \mathfrak{h}^{ab}\Delta_{abcd}^{-1}\mathfrak{h}^{cd} + \mathfrak{h}^{ab}\Delta_{ab}^{-1}\mathcal{K} + \mathcal{K}\Delta_{ab}^{-1}\mathfrak{h}^{ab} + \mathcal{K}\Delta^{-1}\mathcal{K} \right), \tag{2.12}$$

with

$$\Delta^{-1} = -\partial^2 + \mu^2, \tag{2.13a}$$

$$\Delta_{ab}^{-1} = -\eta_{ab}\left(\partial^2 - \frac{1}{2}\mu^2(\lambda - 1)\right) + \partial_a\partial_b, \tag{2.13b}$$

$$\Delta_{abcd}^{-1} = \frac{1}{2}\mu^2 \left(\eta_{ac}x_{[b}\partial_{d]} + \eta_{bd}x_{[a}\partial_{c]} + \eta_{ab}x_{(c}\partial_{d)} - \eta_{cd}x_{(a}\partial_{b)}\right)$$
$$+ \frac{\mu^2(\lambda + 1)}{2}\left(\eta_{ab}\eta_{cd} - \eta_{a(c}\eta_{d)b}\right). \tag{2.13c}$$

The detailed calculation can be found in [4]. Just as in the case of the odd modes, the conformal flatness of the longitudinal metric $g_{ab} = A(x,y)\eta_{ab}$ has been exploited to write the above action. Therefore all indices are raised and lowered with the flat metric.

### 2.0.3 Action of gauge transformations on the graviton

In order to understand the gauge fixing we employ, we first write down the effect of a gauge transformation on the different graviton modes $H_{ab}, h_a^+, h_a^-, G, K, h_\Omega$. These will also come to use in defining ghosts for each partial wave. Let us define the diffeomorphisms $\bar{\xi}^\mu$ along the longtidunal and transverse directions as

$$\bar{\xi}_a = \sum_{\ell,m} \xi_a Y_\ell^m \quad \text{and} \quad \bar{\xi}_A = \sum_{\ell,m} \xi^+ \partial_A Y_\ell^m + \sum_{\ell,m} \xi^- \epsilon_A{}^B \partial_B Y_\ell^m, \tag{2.14}$$

where we split the transverse part into even ($\xi^+$) and odd modes ($\xi^-$). In what follows, $\delta^{(i)}A$ denotes the i-th order variation of the tensor $A$

$$g_{\sigma\rho}\delta^{(1)}\Gamma_{\mu\nu}^\rho = -h_{\sigma\rho}\Gamma_{\mu\nu}^\rho + \frac{1}{2}(\partial_\mu h_{\sigma\nu} + \partial_\nu h_{\sigma\mu} - \partial_\sigma h_{\mu\nu})$$
$$= \frac{1}{2}(\nabla_\mu h_{\sigma\nu} + \nabla_\nu h_{\sigma\mu} - \nabla_\sigma h_{\mu\nu}). \tag{2.15}$$

**The longitudinal modes**  The action of the diffeomorphisms on the longitudinal modes work out to

$$\delta h_{ab} = \left[\partial_a \xi_b + \partial_b \xi_a - 2\Gamma^c_{ab}\xi_c\right] Y^m_\ell - 2\delta\Gamma^\mu_{ab}\bar{\xi}_\mu. \tag{2.16}$$

This shows that

$$\delta^{(1)}H_{ab} = \tilde{\nabla}_a \xi_b + \tilde{\nabla}_b \xi_a, \tag{2.17}$$

$$\delta^{(2)}H_{ab} = \bar{\xi}^c\left(\tilde{\nabla}_c h_{ab} - \tilde{\nabla}_a h_{cb} - \tilde{\nabla}_b h_{ac}\right) - 2\bar{\xi}^A\tilde{\nabla}_{(a}h_{b)A} + \bar{\xi}^A\partial_A h_{ab} =: F_{ab}\left[h,\bar{\xi}\right], \tag{2.18}$$

where the tilde stands for operators that are only defined on the longitudinal directions.[2] The second order term is sub-leading and will be irrelevant for our analysis. Furthermore, since the gauge parameter $\bar{\xi}$ and the graviton are in different partial waves in general, we refrain from further simplification of this term.

**The transverse modes**  The variation of the transverse modes under the said gauge transformations can be written as

$$\begin{aligned}
\delta h_{AB} &= \partial_A\bar{\xi}_B + \partial_B\bar{\xi}_A - 2\Gamma^\mu_{AB}\bar{\xi}_\mu - 2\delta\Gamma^\mu_{AB}\bar{\xi}_\mu \\
&= 2\xi^+\left(\partial_A\partial_B - \Gamma^C_{AB}\partial_C\right)Y^m_\ell + 2\xi^-\left(\partial_{(A}\epsilon_{B)}{}^C\partial_C - \Gamma^C_{AB}\epsilon_C{}^D\partial_D\right)Y^m_\ell \\
&\quad + 2g_{AB}V^a\xi_a Y^m_\ell - 2\delta\Gamma^\mu_{AB}\xi_\mu.
\end{aligned} \tag{2.19}$$

Comparing with the components appearing in $h_{AB}$, we see that

$$\delta^{(1)}G = 2\xi^+, \quad \delta^{(1)}K = 2V^a\xi_a \quad \text{and} \quad \delta^{(1)}h_\Omega = 2\xi^-, \tag{2.20}$$

and

$$\begin{aligned}
\delta^{(2)}h_{AB} &= \bar{\xi}^b\left(-2\tilde{\nabla}_{(A}h_{B)b} - 2V^a h_{ab}g_{AB} + \partial_b h_{AB}\right) \\
&\quad - \bar{\xi}^C\left(\tilde{\nabla}_A h_{CB} + \tilde{\nabla}_B h_{CA} - \tilde{\nabla}_C h_{AB} + 2V^a h_{aC}g_{AB}\right).
\end{aligned}$$

**The mixed modes**  The diffeomorphisms also act on the modes that mix the longitudinal and transverse ones in the following way

$$\delta h_{aA} = \left[(\partial_a - 2V_a)\xi^+ + \xi_a\right]\partial_A Y^m_\ell + (\partial_a - 2V_a)\xi^-\epsilon_A{}^B\partial_B Y^m_\ell - 2\delta\Gamma^\mu_{aA}\xi_\mu. \tag{2.21}$$

This implies that

$$\delta^{(1)}h^+_a = (\partial_a - 2V_a)\xi^+ + \xi_a \quad \text{and} \quad \delta^{(1)}h^-_a = (\partial_a - 2V_a)\xi^-, \tag{2.22}$$

and

$$\delta^{(2)}h_{aA} = \bar{\xi}^b\left(-\partial_A h_{ab} + 2\tilde{\nabla}_{[b}h_{a]A} + 2V_a h_{bA}\right) + \bar{\xi}^B\left(2\tilde{\nabla}_{[B}h_{A]a} + 2V_a h_{AB} - \partial_a h_{AB}\right).$$

---

[2]For instance, this would imply that $\tilde{\nabla}_a h_{bA} = \partial_a h_{bA} - \Gamma^c_{ab}h_{cA}$ as $h_{bA}$ is a vector along the longitudinal directions.

## 2.1 Propagator for the monopole mode: $\ell = 0$

The odd harmonic components, all of which contain angular derivatives, vanish identically for the monopole since the corresponding spherical harmonic is constant. For the same reason, so do the angular diffeomorphisms in (2.14). The non-vanishing even harmonic mode is now

$$h_{00,\mu\nu}^+ = \begin{pmatrix} H_{xx} & H_{xy} & 0 & 0 \\ & H_{yy} & 0 & 0 \\ & & r^2 K & 0 \\ & & & r^2 K \sin^2\theta \end{pmatrix} Y_0, \tag{2.23}$$

and we have the two longitudinal diffeomorphisms in (2.14) to avail. One convenient choice is to set the transverse scalar $K$ to zero. The diffeomorphism that keeps it fixed to zero can be worked out from (2.20) to be one that satisfies

$$2V^a \bar\xi_a = -K Y_0. \tag{2.24}$$

This implies that $\delta K = -K$ ensuring that there are no transverse degrees of freedom left.

The last redundant degree of freedom that is left to be fixed can be chosen to be $h_{xy} = 0$. We will call this the *traceless gauge* because it is equivalent[3] to $\eta^{ab} h_{ab} = 0$. An alternative choice was proposed in [29, 30] $x_a \epsilon_{bc} x^c h^{ab} = 0$, and was called the generalised Regge-Wheeler gauge. As we will find in Section 4, the traceless gauge has great calculational utility, while it was observed in [29, 30] that the generalised Regge-Wheeler gauge makes the working out of ghost actions simpler. The diffeomorphism that fixes the last gauge degree of freedom should not change the transverse mode we already fixed and therefore, it must satisfy the linearly independent relation $V^a \xi_a = 0$. So, a natural guess for the longitudinal diffeomorphism would be

$$\bar\xi_a = -\frac{V_a}{2V^2} K Y_0 + f(x,y) \epsilon_{ab} V^b Y_0. \tag{2.25}$$

To ensure that the second term indeed fixes the trace degree of freedom, we demand that $\delta h_{ab} = -\eta^{ab} \mathfrak{h}_{ab}$ which can be worked out to be

$$(V \cdot \partial) f + f (\nabla \cdot V) = -\eta^{ab} \mathfrak{h}_{ab} =: -\mathfrak{h}. \tag{2.26}$$

In Schwarzschild coordinates, this equation can be rewritten as

$$\partial_r f + \frac{(\nabla \cdot V)}{V^r} f = -\frac{\mathfrak{h}}{V^r}, \tag{2.27}$$

which has a solution with the following integral representation

$$f = -I(r) \int_R^r \frac{\mathfrak{h}(r',t)}{I(r') V^r(r')} dr' \quad \text{with} \quad I(r) = \mathrm{Exp}\left[ -\int_R^r \frac{(\nabla \cdot V)(r')}{V^r(r')} dr' \right]. \tag{2.28}$$

---

[3]Note that the flat metric $\eta_{ab}$ is off-diagonal in the lightcone coordinates we are working in.

### 2.1.1 Traceless gauge

As we have demonstrated above, in the traceless gauge, the only remaining off-shell degrees of freedom are the diagonal longitudinal modes

$$\mathfrak{h}_0^{ab} = \begin{pmatrix} \mathfrak{h}_0^{xx} & 0 \\ 0 & \mathfrak{h}_0^{yy} \end{pmatrix}. \tag{2.29}$$

The quadratic action is

$$S = \frac{1}{4} \int \mathrm{d}^2 x \; \mathfrak{h}_0^{ab} \Delta_{abcd}^{-1} \mathfrak{h}_0^{cd}, \tag{2.30}$$

where the quadratic operator is given by

$$\Delta_{abcd}^{-1} = \frac{1}{2} \mu^2 \left( \eta_{ac} x_{[b} \partial_{d]} + \eta_{bd} x_{[a} \partial_{c]} \right) + \frac{1}{2} \mu^2 \left( \eta_{ab} \eta_{cd} - \eta_{ac} \eta_{bd} - \eta_{ad} \eta_{bc} \right). \tag{2.31}$$

This is the operator that was found in [4] for the even modes, only made traceless (as defined in Appendix B) to fix the redundant gauge degree of freedom. In momentum space the operator reads

$$\Delta_{abcd}^{-1} = \frac{1}{2} \mu^2 \left( \eta_{ac} k_{[b} \partial_{d]}^k + \eta_{bd} k_{[a} \partial_{c]}^k \right) + \frac{1}{2} \mu^2 \left( \eta_{ab} \eta_{cd} - \eta_{ac} \eta_{bd} - \eta_{ad} \eta_{bc} \right) \tag{2.32}$$

where $\partial_a^k$ is the derivative with respect to $k_a$. In order to invert this operator, we begin by observing that time-translational invariance (in Schwarzschild coordinates) or dilation invariance (under $x \to \lambda x$ and $y \to \lambda^{-1} y$ in Kruskal-Szekeres coordinates) implies that the most general form of the inverse is symmetric in the first and last two indices, in addition to being symmetric under exchange of the first two indices with the last two:

$$\begin{aligned} \Delta_{abcd} = {}& A\left(k^2\right) \eta_{ab} \eta_{cd} + B\left(k^2\right) \eta_{a(c} \eta_{d)b} + C\left(k^2\right) \left( \eta_{ab} k_c k_d + \eta_{cd} k_a k_b \right) \\ & + D\left(k^2\right) \left( \eta_{a(c} k_{d)} k_b + \eta_{b(c} k_{d)} k_a \right) + E\left(k^2\right) k_a k_b k_c k_d. \end{aligned} \tag{2.33}$$

In addition, to respect the traceless gauge, we require the inverse to be traceless (as discussed in Appendix B) such that

$$\begin{aligned} \Delta_{abcd}^{-1} \Delta^{cdef} = {}& \delta_a^{(e} \delta_b^{f)} - \frac{1}{2} \eta_{ab} \eta^{ef} \\ = {}& \frac{B\mu^2}{2} \eta_{ab} \eta^{ef} - \left( B\mu^2 + \frac{\mu^2 D k^2}{4} \right) \delta_a^{(e} \delta_b^{f)} + \frac{\mu^2 D}{4} \left( \eta_{ab} k^e k^f + \eta^{ef} k_a k_b \right) \\ & - \left( \frac{\mu^2 D}{2} + \mu^2 E k^2 \right) k_{(a} \delta_{b)}^{(e} k^{f)} + \mu^2 E k_a k_b k^e k^f. \end{aligned} \tag{2.34}$$

where in the second equality, we plugged in[4] the general form (2.33). The solution to this equation is given by

$$B\left(k^2\right) = -\frac{1}{\mu^2} \quad \text{and} \quad D\left(k^2\right) = 0 = E\left(k^2\right), \tag{2.35}$$

---

[4]Notice that $\partial_a^k F\left(k^2\right) = 2F' k_a$ and so on for other similar terms. This implies that $k_{[a} \partial_{b]}^k F\left(k^2\right) = 2F' k_{[a} k_{b]} = 0$ allowing us to drop all derivative terms acting on scalar functions.

with arbitrary functions $A\left(k^2\right)$ and $C\left(k^2\right)$. These are determined by demanding the inverse $\Delta_{abcd}$ to be traceless and are given by

$$A\left(k^2\right) \;=\; \frac{1}{2\mu^2} \quad \text{and} \quad C\left(k^2\right) \;=\; 0\,. \tag{2.36}$$

Therefore, the propagator is given by

$$\mathscr{P}_{\ell=0}^{abcd} \;=\; \frac{1}{2\mu^2}\left(\eta^{ab}\eta^{cd} - \eta^{ac}\eta^{bd} - \eta^{ad}\eta^{bc}\right). \tag{2.37}$$

Evidently, the propagator has no momentum dependence. This is owed to the fact that there are only two off-shell degrees of freedom in the monopole and therefore, imposing equations of motion, we are left with no dynamical modes.

### 2.1.2 'Generalised Regge-Wheeler' gauge

Instead of the traceless gauge, had we used the generalised Regge-Wheeler gauge $x_a x^b \epsilon_{bc} h^{ac} = 0$ proposed in [29], the quadratic operator turn out to be the same as those found in [4]

$$\Delta_{abcd}^{-1} \;=\; \frac{1}{2}\mu^2\left(\eta_{ac}x_{[b}\partial_{d]} + \eta_{bd}x_{[a}\partial_{c]} + \eta_{ab}x_{(c}\partial_{d)} - \eta_{cd}x_{(a}\partial_{b)}\right) \tag{2.38}$$

$$+\, \mu^2\left(\eta_{ab}\eta_{cd} - \eta_{a(c}\eta_{d)b}\right), \tag{2.39}$$

and the propagator is readily found to be the same as the one derived in [4]

$$\mathscr{P}_{\ell=0}^{abcd} \;=\; \frac{1}{2\mu^2}\left(2\eta^{ab}\eta^{cd} - \eta^{ac}\eta^{bd} - \eta^{ad}\eta^{bc}\right). \tag{2.40}$$

While the propagator is still non-dynamical, it does contain the trace mode. This trace can be separated out as $\mathfrak{h}_{ab} = \hat{\mathfrak{h}}_{ab} + \frac{1}{2}\eta_{ab}\mathfrak{h}$. As we will see in Section 4, depending on the form of the interactions allowed in the theory, the trace mode can sometimes be integrated out and the two gauge choices we discussed coincide.

## 2.2 Propagator for the dipole mode: $\ell = 1$

Before fixing any gauge, the dipole mode $\ell = 1$ contains the following graviton degrees of freedom, as can be checked by explicit calculation for each of the allowed values foor $m = -1, 0, 1$.

$$h_{1m,\mu\nu}^{-} \;=\; \begin{pmatrix} 0 & 0 & -h_x^{-}\csc\theta\partial_\phi & h_x^{-}\sin\theta\partial_\theta \\ & 0 & -h_y^{-}\csc\theta\partial_\phi & h_y^{-}\sin\theta\partial_\theta \\ & & 0 & 0 \\ & & & 0 \end{pmatrix} Y_1^m\,, \tag{2.41}$$

$$h_{1m,\mu\nu}^{+} \;=\; \begin{pmatrix} H_{xx} & H_{xy} & h_x^{+}\partial_\theta & h_x^{+}\partial_\phi \\ & H_{yy} & h_y^{+}\partial_\theta & h_y^{+}\partial_\phi \\ & & r^2(K-G) & 0 \\ & & & r^2(K-G)\sin^2\theta \end{pmatrix} Y_1^m\,. \tag{2.42}$$

### 2.2.1 Propagator for the even harmonics

We see from (2.42) that the only transverse scalar mode is $K - G$ which we may conveniently just call $K$. Therefore, we have

$$
h^+_{1m,\mu\nu} = \begin{pmatrix} H_{xx} & H_{xy} & h^+_x \partial_\theta & h^+_x \partial_\phi \\ & H_{yy} & h^+_y \partial_\theta & h^+_y \partial_\phi \\ & & r^2 K & 0 \\ & & & r^2 K \sin^2\theta \end{pmatrix} Y^m_1 . \tag{2.43}
$$

Of the four gauge degrees of freedom to be fixed, three are even modes. The convenient choice would be to gauge fix the transverse scalar and $h^+_a$ which we do by the following choice

$$
\xi_a = \left( \frac{1}{2} r^2 \partial_a K - h^+_a \right) Y^m_1 \quad \text{and} \quad \xi_A = -\frac{1}{2} r^2 K \partial_A Y^m_1 , \tag{2.44}
$$

to reduce the even mode to

$$
h^+_{1m,\mu\nu} = \begin{pmatrix} H_{xx} & H_{xy} & 0 & 0 \\ & H_{yy} & 0 & 0 \\ & & 0 & 0 \\ & & & 0 \end{pmatrix} Y^m_1 . \tag{2.45}
$$

The remaining degree of freedom must be removed from the odd mode. The quadratic action for the even wave is therefore

$$
S = \sum_{m=-1}^{1} \frac{1}{4} \int d^2x \; \mathfrak{h}^{ab}_{1m} \Delta^{-1}_{abcd} \mathfrak{h}^{cd}_{1m} , \tag{2.46}
$$

where

$$
\Delta^{-1}_{abcd} = \frac{\mu^2}{2} \left( \eta_{ac} x_{[b} \partial_{d]} + \eta_{bd} x_{[a} \partial_{c]} + \eta_{ab} x_{(c} \partial_{d)} - \eta_{cd} x_{(a} \partial_{b)} \right) + 2\mu^2 \left( \eta_{ab} \eta_{cd} - \eta_{a(c} \eta_{d)b} \right) . \tag{2.47}
$$

To find the propagator, we first observe that the even mode carries three offshell degrees of freedom and the odd mode comes with one (after gauge fixing the other). This implies that upon imposing equations of motion, there are no dynamical degrees of freedom left. So, we expect that the derivative terms in the quadratic operator will not contribute to the propagator. So, we may use the general ansatz (2.33) but without requiring tracelessness. Therefore, we find

$$
\begin{aligned}
\Delta^{-1}_{abcd} \Delta^{cdef} &= \delta^{(e}_a \delta^{f)}_b \\
&= 2\mu^2 (A + B) \eta_{ab} \eta^{ef} - 2\mu^2 B \delta^{(e}_a \delta^{f)}_b + \text{derivative terms} .
\end{aligned} \tag{2.48}
$$

where we used the momentum space representation ($x \to i\partial^k$ and $\partial \to -ik$) in the second equality. The solution is found to be

$$
A(k^2) = \frac{1}{2\mu^2} , \quad B(k^2) = -\frac{1}{2\mu^2} \quad \text{and} \quad C(k^2) = D(k^2) = E(k^2) = 0 . \tag{2.49}
$$

So, the propagator is given by

$$\mathscr{P}_{1m}^{abcd} = \frac{1}{4\mu^2} \left( 2\eta^{ab}\eta^{cd} - \eta^{ac}\eta^{bd} - \eta^{ad}\eta^{bc} \right). \tag{2.50}$$

### 2.2.2 Propagator for the odd harmonics

The action for the odd dipole mode can be read off from the general near-horizon action (C.24)

$$S_{1,m} = -\int \mathrm{d}^2x \; \mathfrak{h}^a \left( \partial_a \partial_b - \eta_{ab}\partial^2 - \frac{3}{2}\mu^2 x_a \partial_b + \frac{3}{2}\mu^2 x_b \partial_a + \frac{3}{2}\mu^2 \eta_{ab} \right) \mathfrak{h}^b. \tag{2.51}$$

A convenient choice of gauge to fix the redundant degree of freedom in the odd mode $h^a$ is a Lorenz-like gauge since it is a spin one field in question. We choose $\partial_a h^a = 0$ whose corresponding diffeomorphism is given by $\xi_A = -F(x)\,\epsilon_A{}^B\partial_B$, where $F$ is given by a solution to

$$\partial^2 F(x) - 2V^a\partial_a F(x) - 2F(x)\partial_a V^a = -\partial_a \mathfrak{h}^a. \tag{2.52}$$

Owing to the gauge choice, the first and third terms in the quadratic operator vanish identically. The fourth term can be integrated by parts as

$$\int \mathrm{d}^2x \; \mathfrak{h}^a x_b \partial_a \mathfrak{h}^b = \int \mathrm{d}^2x \; \mathfrak{h}^a \partial_a \left( x_b \mathfrak{h}^b \right) - \int \mathrm{d}^2x \; \mathfrak{h}^a \eta_{ab} \mathfrak{h}^b$$

$$= -\int \mathrm{d}^2x \; \mathfrak{h}^b x_b \partial_a \mathfrak{h}^a - \int \mathrm{d}^2x \; \mathfrak{h}^a \eta_{ab} \mathfrak{h}^b, \tag{2.53}$$

where the first integral drops due to the gauge choice and the second term cancels the last term in (2.51), leaving us with

$$S_{1,m} = \int \mathrm{d}^2x \; \mathfrak{h}^a \eta_{ab} \partial^2 \mathfrak{h}^b. \tag{2.54}$$

The propagator is therefore easily found to be the familiar one for a massless spin one field

$$\mathscr{P}_{1m}^{ab} = \frac{\eta^{ab}}{k^2 - i\epsilon}. \tag{2.55}$$

In the generalised Regge-Wheeler gauge of [29, 30], $V^a h_a = 0$, the quadratic operator takes the following non-invertible form: $\partial_a \partial_b - \partial^2 \eta_{ab}$. However, defining $h^a = h_-\epsilon^{ab}V_b$ results in a propagator for the field $h_-$. It would be interesting to understand the apparent gauge-dependent emergent dynamics for what is arguably a non-propagating mode [29, 30].

### 2.3 Propagator for the multipole modes: $\ell > 1$

All ten components of the graviton are non-trivial for the multipole modes. So, the gauge choice we employed in [4] to fix four of those was

$$\xi_a = \left( \frac{1}{2}r^2\partial_a G - h_a^+ \right) Y_\ell^m \quad \text{and} \quad \xi_A = -\frac{1}{2}r^2 G \partial_A Y_\ell^m - \frac{1}{2}h_\Omega \epsilon_A{}^B \partial_B Y_\ell^m. \tag{2.56}$$

This diffeomorphism sets $h_a^+ = G = h_\Omega = 0$, resulting in the following graviton modes

$$h_{\ell m,\mu\nu}^- = \begin{pmatrix} 0 & 0 & -h_x \csc\theta\partial_\phi & h_x \sin\theta\partial_\theta \\ 0 & 0 & -h_y \csc\theta\partial_\phi & h_y \sin\theta\partial_\theta \\ -h_x \csc\theta\partial_\phi & -h_y \csc\theta\partial_\phi & 0 & 0 \\ h_x \sin\theta\partial_\theta & h_y \sin\theta\partial_\theta & 0 & 0 \end{pmatrix} Y_\ell^m \qquad (2.57)$$

$$h_{\ell m,\mu\nu}^+ = \begin{pmatrix} H_{xx} & H_{xy} & 0 & 0 \\ H_{xy} & H_{yy} & 0 & 0 \\ 0 & 0 & r^2 K & 0 \\ 0 & 0 & 0 & r^2 K \sin^2\theta. \end{pmatrix} Y_\ell^m . \qquad (2.58)$$

### 2.3.1 Propagator for the even harmonics

The near-horizon propagator for the even multipole modes was derived in full generality in [3, 4], to which we refer for details. Here, we collect the results from those references

$$S = \sum_{\ell m} \frac{1}{4} \int \mathrm{d}^2 x \left( \mathfrak{h}^{ab} \Delta_{abcd}^{-1} \mathfrak{h}^{cd} + \mathfrak{h}^{ab} \Delta_{ab}^{-1} \mathcal{K} + \mathcal{K} \Delta_{ab}^{-1} \mathfrak{h}^{ab} + \mathcal{K} \Delta^{-1} \mathcal{K} \right), \qquad (2.59)$$

with

$$\Delta_{abcd}^{-1} = \frac{\mu^2}{2} \left( \eta_{ac} x_{[b}\partial_{d]} + \eta_{bd} x_{[a}\partial_{c]} + \eta_{ab} x_{(c}\partial_{d)} - \eta_{cd} x_{(a}\partial_{b)} \right) + \frac{\mu^2(\lambda+1)}{2} \left( \eta_{ab}\eta_{cd} - \eta_{a(c}\eta_{d)b} \right)$$

$$\Delta_{ab}^{-1} = -\eta_{ab} \left( \partial^2 - \frac{1}{2}\mu^2(\lambda-1) \right) + \partial_a\partial_b ,$$

$$\Delta^{-1} = -\partial^2 + \mu^2 . \qquad (2.60)$$

The propagators are given by compositions of the inverses of the above operators:

$$\mathscr{P}_{abcd} = \Delta_{abcd} + \mathscr{P}_{\mathcal{K}} \mathrm{p}_{ab}\mathrm{p}_{cd} \qquad (2.61)$$

$$\mathscr{P}_{ab} = -\mathscr{P}_{\mathcal{K}} \mathrm{p}_{ab} \qquad (2.62)$$

$$\mathscr{P}_{\mathcal{K}} = \frac{\lambda+1}{\lambda-3} \frac{1}{k^2 + \mu^2\lambda} , \qquad (2.63)$$

where

$$\Delta_{abcd} = \frac{1}{\mu^2(\lambda+1)} \left( 2\eta_{ab}\eta_{cd} - \eta_{ac}\eta_{bd} - \eta_{ad}\eta_{bc} \right) \qquad (2.64)$$

$$\mathrm{p}_{ab} = \frac{\lambda-1}{\lambda+1}\eta_{ab} + \frac{2k_a k_b}{\mu^2(\lambda+1)} . \qquad (2.65)$$

Asymptotically, the propagator scales as $k^2$, a result of the effective mass of the two-dimensional graviton. As was also pointed out in [4], the shape of the propagators resembles that of massive gravity, with mass $\mu^2(\lambda+1)$. Moreover, there is a change in sign for the $\ell = 0$ monopole and a pole in the $\ell = 1$ dipole mode, indicating additional redundant gauge degrees of freedom. These were resolved in the previous subsections 2.1 and 2.2 to find the corresponding propagators. Therefore, the above propagators are valid strictly for the $\ell > 1$ multipole modes.

### 2.3.2 Propagator for the odd harmonics

The near-horizon action for the odd harmonics of the multipole modes is derived in detail in Appendix C. The result is (C.24) which we repeat here:

$$\sum_{\ell,m} S_{\ell,m} = -\sum_{\ell,m} \frac{\lambda-1}{2} \int \mathrm{d}^2x \, \mathfrak{h}^a \left( \partial_a \partial_b - \eta_{ab} \partial^2 - 3\mu^2 x_{[a} \partial_{b]} + \mu^2 \eta_{ab} \left( \lambda - \frac{3}{2} \right) \right) \mathfrak{h}^b . \quad (2.66)$$

The propagator is then found to be (C.34):

$$\mathcal{G}^{ab} = \frac{\lambda - \frac{9}{2}}{\lambda - 1} \frac{1}{k^2 \left( \lambda - 3 \right) + \mu^2 \left( \lambda - \frac{3}{2} \right) \left( \lambda - \frac{9}{2} \right)} \left( \eta^{ab} + \frac{k^a k^b}{\mu^2 \left( \lambda - \frac{9}{2} \right)} \right) . \quad (2.67)$$

This shape of this propagator resembles that of a massive spin-1 particle whose mass diverges when $\lambda = 3$ or $\ell = 1$. This is not a physical pole in momentum space and suggests additional gauge redundancy in that dipole mode which warranted a separate study of its propagator in Section 2.2 where we resolved this issue. Therefore, the propagator above is only valid and reliable for the multipole modes $\ell \geq 2$.

## 3 Interactions with matter

In this section, we will proceed to (minimally) couple the theory we have considered so far, to matter (with an action we label $S_M$). The leading interaction term takes the form

$$S_{int} = \frac{1}{2}\kappa \int \mathrm{d}^4x \sqrt{-g} \, h^{\mu\nu} T_{\mu\nu} \quad \text{with} \quad T_{\mu\nu} := \frac{-2}{\sqrt{-g}} \frac{\delta S_M}{\delta g^{\mu\nu}} . \quad (3.1)$$

For the even modes, using (2.58), we have

$$\begin{aligned}
S_{int}^+ &= \sum_{\ell m} \frac{1}{2}\kappa \int \mathrm{d}^2x \left( r^2 A\left(x,y\right) \right) \mathrm{d}\Omega_2 \left( g^{ac} g^{bd} H_{cd} T_{ab} + K g^{AB} T_{AB} \right) Y_\ell^m \\
&= \sum_{\ell m} \frac{1}{2}\kappa \int \mathrm{d}^2x \left( r^2 A\left(x,y\right) \right) \mathrm{d}\Omega_2 \left( \eta^{ac} \eta^{bd} \frac{A\left(x,y\right)}{r A\left(x,y\right)^2} \mathfrak{h}_{cd} T_{ab} + \frac{\mathcal{K}}{r} g^{AB} T_{AB} \right) Y_\ell^m \\
&= \sum_{\ell m} \frac{1}{2}\kappa \int \mathrm{d}^2x \, r \, \mathrm{d}\Omega_2 \left( \mathfrak{h}^{ab} T_{ab} + \mathcal{K} g^{AB} T_{AB} \right) Y_\ell^m ,
\end{aligned} \quad (3.2)$$

where in the second line, we used the field definitions $rH_{ab} = A\left(x,y\right) \mathfrak{h}_{ab}$, $rK = \mathcal{K}$ and the fact that $A\left(x,y\right) g^{ab} = \eta^{ab}$. Considering a minimally coupled scalar field, we have

$$T_{\mu\nu} = \partial_\mu \tilde{\phi} \partial_\nu \tilde{\phi} - \frac{1}{2} g_{\mu\nu} \partial_\rho \tilde{\phi} \partial^\rho \tilde{\phi} . \quad (3.3)$$

Therefore, we have

$$
\begin{aligned}
T_{ab} &= \partial_a\tilde{\phi}\partial_b\tilde{\phi} - \frac{1}{2}g_{ab}\partial_\rho\tilde{\phi}\partial^\rho\tilde{\phi} \\
&= \left(\frac{1}{r}\partial_a\phi + \phi\partial_a\frac{1}{r}\right)\left(\frac{1}{r}\partial_b\phi + \phi\partial_b\frac{1}{r}\right) - \frac{1}{2}g_{ab}\left(\frac{1}{r}\partial_\rho\phi + \phi\partial_\rho\frac{1}{r}\right)\left(\frac{1}{r}\partial^\rho\phi + \phi\partial^\rho\frac{1}{r}\right) \\
&= \frac{1}{r^2}\left(\partial_a\phi\partial_b\phi - \frac{1}{2}\eta_{ab}\eta^{cd}\partial_c\phi\partial_d\phi - \frac{\eta_{ab}}{2A\left(x,y\right)r^2}\gamma^{CD}\partial_C\phi\partial_D\phi\right) + \dots \\
&\sim \frac{1}{R^2}\left(\partial_a\phi\partial_b\phi - \frac{1}{2}\eta_{ab}\eta^{cd}\partial_c\phi\partial_d\phi - \frac{1}{2}\eta_{ab}\gamma^{CD}\hat{\partial}_C\phi\hat{\partial}_D\phi\right) \\
&\sim \frac{1}{R^2}\left(\partial_a\phi\partial_b\phi - \frac{1}{2}\eta_{ab}\eta^{cd}\partial_c\phi\partial_d\phi\right) =: \frac{1}{R^2}\tilde{T}_{ab}\,,
\end{aligned}
\tag{3.4}
$$

where we redefined the scalar field as $\tilde{\phi} \to \frac{1}{r}\phi$ in the second line. The terms represented by the dots evidently drop out in the near-horizon limit $r \sim R$, $A\left(x,y\right) \sim 1$. In the third line, we also used that $g^{CD} = \frac{1}{r^2}\gamma^{CD}$ where $\gamma^{CD}$ is the inverse of the round metric on the unit two-sphere. In the fourth line, we defined the rescaled transverse derivatives $\hat{\partial}_A := \frac{1}{R}\partial_A$ in the near horizon limit. The transverse effects are, therefore, naturally suppressed in the near-horizon limit by an additional $\frac{1}{R}$ factor in comparison to the longitudinal momenta. Therefore, in the fifth line, we ignore the last term of the fourth line and arrive at the energy momentum tensor purely along the longitudinal directions.

Similarly, for the transverse and mixed components of the stress tensor, we have

$$
\begin{aligned}
g^{AB}T_{AB} &= g^{AB}\partial_A\tilde{\phi}\partial_B\tilde{\phi} - g^{ab}\partial_a\tilde{\phi}\partial_b\tilde{\phi} - g^{CD}\partial_C\tilde{\phi}\partial_D\tilde{\phi} \\
&= -\frac{1}{A\left(x,y\right)}\eta^{ab}\left(\frac{1}{r}\partial_a\phi + \phi\partial_a\frac{1}{r}\right)\left(\frac{1}{r}\partial_b\phi + \phi\partial_b\frac{1}{r}\right) \\
&\sim -\frac{1}{R^2}\eta^{ab}\partial_a\phi\partial_b\phi
\end{aligned}
\tag{3.5}
$$

$$
T_{aA} = \partial_a\tilde{\phi}\partial_A\tilde{\phi} \sim \frac{1}{R}\partial_a\phi\hat{\partial}_A\phi\,.
\tag{3.6}
$$

Therefore, in the near-horizon limit, the interaction term for the even graviton takes the following form:

$$
\begin{aligned}
S_{int}^+ &= \frac{\kappa}{2R}\sum_{\substack{\ell m \\ \ell_1 m \\ \ell_2 m}}\int \mathrm{d}^2x\,\mathrm{d}\Omega_2\left[\mathfrak{h}^{ab}\left(\partial_a\phi_{\ell_1,m_1}\partial_b\phi_{\ell_2,m_2} - \frac{1}{2}\eta_{ab}\partial_c\phi_{\ell_1,m_1}\partial^c\phi_{\ell_2,m_2}\right)\right. \\
&\qquad\qquad\qquad\qquad\qquad\left. - \mathscr{K}\eta^{ab}\partial_a\phi_{\ell_1,m_1}\partial_b\phi_{\ell_2,m_2}\right]Y_\ell^m Y_{\ell_1}^{m_1}Y_{\ell_2}^{m_2} \\
&= \frac{\gamma}{2}\sum_{\substack{\ell m \\ \ell_1 m \\ \ell_2 m}}\int \mathrm{d}^2x\left[\left(\mathfrak{h}^{ab} - \frac{1}{2}\mathfrak{h}\eta^{ab} - \mathscr{K}\eta^{ab}\right)\partial_a\phi_{\ell_1,m_1}\partial_b\phi_{\ell_2,m_2}\right]C\left[\ell m;\ell_1 m_1;\ell_2 m_2\right]\,,
\end{aligned}
\tag{3.7}
$$

where we defined the coupling constant $\gamma := \kappa/R$ and

$$C\left[\ell m; \ell_1 m_1; \ell_2 m_2\right] \; := \; \int \mathrm{d}\Omega_2 \; Y_\ell^m Y_{\ell_1}^{m_1} Y_{\ell_2}^{m_2} \,. \tag{3.8}$$

An interesting observation to be made from the form of the longitudinal components is that $\eta^{ab}\tilde{T}_{ab} = 0$ provided transverse momenta are small. This implies that the trace mode in the graviton $\mathfrak{h}_{ab}$ does not couple to the matter fields. This allows us to combine the traceless longitudinal graviton and the transverse scalar into a single tensorial mode

$$\hat{\mathfrak{h}}_{ab} \; := \; \mathfrak{h}_{ab} - \frac{1}{2}\eta_{ab}\mathfrak{h} - \mathcal{K}\eta^{ab} \,, \tag{3.9}$$

that couples to the scalar fields as

$$S_{int}^+ \; = \; \frac{\gamma}{2}\sum_{\substack{\ell m \\ \ell_1 m_1 \\ \ell_2 m_2}}\int \mathrm{d}^2 x \left[\hat{\mathfrak{h}}^{ab}\partial_a \phi_{\ell_1,m_1}\partial_b \phi_{\ell_2,m_2}\right] C\left[\ell m; \ell_1 m_1; \ell_2 m_2\right]\,, \tag{3.10}$$

where we suppressed the $\ell, m$ indices on the graviton mode. Owing to residual gauge redundancy that needs to be fixed, $\mathfrak{h}$ and $\mathcal{K}$ vanish identically in the monopole ($\ell = 0$) mode whereas only the latter vanishes for the dipole ($\ell = 1$) mode as discussed in Section 2.1 and Section 2.2, respectively. For the multipole modes ($\ell > 1$), the trace of (3.9) shows that $\mathcal{K} = \frac{1}{2}\hat{\mathfrak{h}}$. In momentum space, the above interaction action can be written as

$$S_{int}^+ \; = \; \frac{\gamma}{2\left(2\pi\right)^6}\sum_{\substack{\ell m \\ \ell_1 m_1 \\ \ell_2 m_2}}\int \mathrm{d}^2 x \int \mathrm{d}^2 p \int \prod_{i=1}^2 \mathrm{d}^2 p_i \left[\hat{\mathfrak{h}}^{ab}\left(p\right) p_{1,a} p_{2,b}\right] C\left[\ell m; \ell_1 m_1; \ell_2 m_2\right]$$

$$\times \phi_{\ell_1,m_1}\left(p_1\right)\phi_{\ell_2,m_2}\left(p_2\right) e^{i(p+p_1+p_2)x}$$

$$= \; \frac{\gamma}{2\left(2\pi\right)^4}\sum_{\substack{\ell m \\ \ell_1 m_1 \\ \ell_2 m_2}}\int \mathrm{d}^2 p \int \prod_{i=1}^2 \mathrm{d}^2 p_i \left[\hat{\mathfrak{h}}^{ab}\left(p\right) p_{1,a} p_{2,b}\right] C\left[\ell m; \ell_1 m_1; \ell_2 m_2\right]$$

$$\times \phi_{\ell_1,m_1}\left(p_1\right)\phi_{\ell_2,m_2}\left(p_2\right)\delta^{(2)}\left(p + p_1 + p_2\right)$$

$$= \; \frac{\gamma}{2}\sum_{\substack{\ell m \\ \ell_1 m_1 \\ \ell_2 m_2}}\int \frac{\mathrm{d}^2 p}{\left(2\pi\right)^2}\int \frac{\mathrm{d}^2 p_1}{\left(2\pi\right)^2}\left[\hat{\mathfrak{h}}^{ab}\left(-p\right) p_{1,a}\left(p - p_1\right)_b\right] C\left[\ell m; \ell_1 m_1; \ell_2 m_2\right]$$

$$\times \phi_{\ell_1,m_1}\left(p_1\right)\phi_{\ell_2,m_2}\left(p - p_1\right)$$

$$=: \; \frac{\gamma}{2}\sum_{\ell m}\int \frac{\mathrm{d}^2 p}{\left(2\pi\right)^2}\, \hat{\mathfrak{h}}_{\ell m}^{ab}\left(-p\right)\tilde{T}_{ab}^{\ell m}\left(p\right)\,, \tag{3.11}$$

where we defined

$$
\tilde{T}_{ab}^{\ell m}\,(p) \;\; = \;\; \sum_{\substack{\ell_1 m_1 \\ \ell_2 m_2}} \int \frac{\mathrm{d}^2 p_1}{(2\pi)^2}\; p_{1,a}\,(p-p_1)_b\;\; C\left[\ell m; \ell_1 m_1; \ell_2 m_2\right]
$$

$$
\times\, \phi_{\ell_1, m_1}\,(p_1)\, \phi_{\ell_2, m_2}\,(p-p_1)\,. \tag{3.12}
$$

## 4  Decoupling the non-interacting graviton modes

The interacting dynamical graviton is a linear combination of the traceless longitudinal mode and the transverse scalar, as can be seen from (3.9) and (3.10). Therefore, it is desirable to rewrite the quadratic action (2.59) for the traceless mode $\hat{\mathfrak{h}}_{ab}$. To this end, let us first write the field redefinition as

$$
\begin{pmatrix} \hat{\mathfrak{h}}_{ab} \\ \hat{\mathcal{K}} \end{pmatrix} \;\; := \;\; \hat{\Lambda} \begin{pmatrix} \mathfrak{h}_{cd} \\ \mathcal{K} \end{pmatrix} \;\; = \;\; \begin{pmatrix} \delta_{ab}^{cd} - \frac{1}{2}\eta_{ab}\eta^{cd} & -\eta_{ab} \\ \eta^{cd} & \Lambda \end{pmatrix} \begin{pmatrix} \mathfrak{h}_{cd} \\ \mathcal{K} \end{pmatrix} \;\; = \;\; \begin{pmatrix} \mathfrak{h}_{ab} - \left(\frac{1}{2}\mathfrak{h} + \mathcal{K}\right)\eta_{ab} \\ \mathfrak{h} + \Lambda\mathcal{K} \end{pmatrix}. \tag{4.1}
$$

For the rest of this section, working in momentum space turns out to be more convenient where the above field redefinition remains and the momentum space representation of $\Lambda$ is given by

$$
\Lambda \;\; = \;\; \frac{2}{\mu^2\,(\lambda+1)}\,\left(k^2 + \mu^2\,(\lambda-1)\right). \tag{4.2}
$$

The inverse of the redefinition matrix $\hat{\Lambda}$ can easily be found

$$
\hat{\Lambda}^{-1} \;\; = \;\; \begin{pmatrix} \delta_{ab}^{cd} - \frac{1}{2}\eta_{ab}\eta^{cd} + \frac{1}{4}\Lambda\eta_{ab}\eta^{cd} & \frac{1}{2}\eta_{ab} \\ -\frac{1}{2}\eta^{cd} & 0 \end{pmatrix}. \tag{4.3}
$$

In matrix form, the Lagrangian in (2.59) can be written as

$$
\mathcal{L} \;\; = \;\; \frac{1}{4}\begin{pmatrix} \mathfrak{h}_{ab} & \mathcal{K} \end{pmatrix} \begin{pmatrix} (\Delta^{-1})^{abcd} & (\Delta^{-1})^{ab} \\ (\Delta^{-1})^{cd} & \Delta^{-1} \end{pmatrix} \begin{pmatrix} \mathfrak{h}_{cd} \\ \mathcal{K} \end{pmatrix} \;\; =: \;\; \frac{1}{4}\begin{pmatrix} \mathfrak{h}_{ab} & \mathcal{K} \end{pmatrix} \boldsymbol{\Delta}^{-1} \begin{pmatrix} \mathfrak{h}_{cd} \\ \mathcal{K} \end{pmatrix}. \tag{4.4}
$$

The propagators of the theory are defined by the inverse $\boldsymbol{\Delta}$. The action of the field redefinition (4.1) can be absorbed into a redefinition of the quadratic operator and its inverse as

$$
\begin{pmatrix} \left(\hat{\mathscr{P}}^{-1}\right)^{abcd} & 0 \\ 0 & \hat{\mathscr{P}}^{-1} \end{pmatrix} \;\; = \;\; \left(\hat{\Lambda}^{-1}\right)^T \boldsymbol{\Delta}^{-1}\hat{\Lambda}^{-1} \tag{4.5}
$$

$$
\begin{pmatrix} \hat{\mathscr{P}}^{abcd} & 0 \\ 0 & \hat{\mathscr{P}} \end{pmatrix} \;\; = \;\; \hat{\Lambda}\boldsymbol{\Delta}\hat{\Lambda}^T. \tag{4.6}
$$

This reduces the action (2.59) to its diagonal degrees of freedom. In momentum space, it reads

$$
S \;\; = \;\; \sum_{\ell m} \frac{1}{4} \int \frac{\mathrm{d}^2 p}{(2\pi)^2}\left(\hat{\mathfrak{h}}_{ab}\left(\hat{\mathscr{P}}^{-1}\right)^{abcd}\hat{\mathfrak{h}}_{cd} + \hat{\mathcal{K}}\hat{\mathscr{P}}^{-1}\hat{\mathcal{K}}\right), \tag{4.7}
$$

where the propagators arising from the inverses of the above quadratic operators are

$$\hat{\mathscr{P}}^{abcd} = \frac{1}{\mu^2(\lambda+1)}\left(\eta_{ab}\eta_{cd}-\eta_{ac}\eta_{bd}-\eta_{ad}\eta_{bc}\right)-\frac{\lambda+1}{\lambda-3}\frac{1}{k^2+\mu^2\lambda}\left(\eta_{ab}+\tilde{\mathrm{p}}_{ab}\right)\left(\eta_{cd}+\tilde{\mathrm{p}}_{cd}\right) \quad (4.8)$$

$$\hat{\mathscr{P}} = \frac{4}{\mu^2(\lambda+1)}, \quad (4.9)$$

where

$$\tilde{\mathrm{p}}_{ab} = \frac{2}{\mu^2(\lambda+1)}\left(k_ak_b-\frac{1}{2}k^2\eta_{ab}\right). \quad (4.10)$$

These expressions are strictly valid only for the multipole modes $\ell > 1$. However, by integrating out the non-interacting graviton modes, in what follows, we will find a scalar rewriting of the theory that is valid for all $\ell$. To this end, as we saw in (3.10), the field $\hat{\mathscr{K}}$ is non-interacting and can therefore be integrated out. The position space field redefinition for this freely propagating mode is

$$\hat{\mathscr{K}} = \mathfrak{h} - \frac{2}{\mu^2(\lambda+1)}\left(-\partial^2+\mu^2(\lambda-1)\right)\mathscr{K}, \quad (4.11)$$

and contains derivatives. Therefore, the Gaussian integral appears to contain higher derivatives. However, given that the original theory we began with was a two-derivative Einstein-Hilbert action, this is merely an artefact of the field redefinitions and does not contribute to the physical S-matrix. It is worth emphasising that this decoupling is only a feature of the cubic order of interactions we consider in this article. Henceforth, we will ignore this free scalar mode and focus on the interacting mode $\hat{\mathfrak{h}}_{ab}$. The complete action in momentum space is now

$$\sum_{\ell m}S_{\ell m} = \sum_{\ell m}\frac{1}{4}\int\frac{\mathrm{d}^2p}{(2\pi)^2}\left(\hat{\mathfrak{h}}^{\ell m}_{ab}(-p)\left(\hat{\mathscr{P}}^{-1}\right)^{abcd}(p)\,\hat{\mathfrak{h}}^{\ell m}_{cd}(p)\right)$$
$$-\sum_{\ell m}\frac{1}{2}\int\frac{\mathrm{d}^2p}{(2\pi)^2}\,\phi_{\ell,m}\left(p^2+\mu^2\lambda\right)\phi_{\ell,m}+\sum_{\ell m}\frac{\gamma}{2}\int\frac{\mathrm{d}^2p}{(2\pi)^2}\left[\hat{\mathfrak{h}}^{ab}_{\ell m}(-p)\,\tilde{T}^{\ell m}_{ab}(p)\right], \quad (4.12)$$

where we took the matter action to be that of a minimally coupled scalar field as in [4].[5] As before, $\lambda=\ell^2+\ell+1$. The stress tensor is a convolution in momentum space

$$\tilde{T}^{\ell m}_{ab}(p) := \sum_{\substack{\ell_1m_1\\\ell_2m_2}}\int\frac{\mathrm{d}^2p_1}{(2\pi)^2}\,(p_{1,a})\,(p-p_1)_b\,\phi_{\ell_1,m_1}(p_1)\,\phi_{\ell_2,m_2}(p-p_1)\,C\left[\ell m;\ell_1m_1;\ell_2m_2\right]$$
$$= \sum_{\substack{\ell_1m_1\\\ell_2m_2}}C\left[\ell m;\ell_1m_1;\ell_2m_2\right]\left(\partial_a\phi_{\ell_1m_1}\star\partial_b\phi_{\ell_2m_2}\right). \quad (4.13)$$

---

[5]See Section 4.3 of [4].

To simplify this action, we first transform the graviton mode as

$$\hat{\mathfrak{h}}^{\ell m}_{ab}(-p) \to \hat{\mathfrak{h}}^{\ell m}_{ab}(-p) - \gamma \hat{\mathscr{P}}_{abcd}(p) \tilde{T}^{cd}_{\ell m}(p) \,, \tag{4.14}$$

to find the following action

$$\sum_{\ell m} S_{\ell m} = \frac{1}{4} \sum_{\ell m} \int \frac{\mathrm{d}^2 p}{(2\pi)^2} \left( \hat{\mathfrak{h}}^{\ell m}_{ab}(-p) \hat{\mathscr{P}}^{abcd}(p) \hat{\mathfrak{h}}^{\ell m}_{cd}(p) \right) - \frac{1}{2} \sum_{\ell m} \int \frac{\mathrm{d}^2 p}{(2\pi)^2} \, \phi_{\ell,m} \left( p^2 + \mu^2 \lambda \right) \phi_{\ell,m}$$

$$- \sum_{\ell m} \frac{\gamma^2}{4} \int \frac{\mathrm{d}^2 p}{(2\pi)^2} \left[ \tilde{T}^{\ell m}_{ab}(-p) \hat{\mathscr{P}}^{abcd}(p) \tilde{T}^{\ell m}_{cd}(p) \right] . \tag{4.15}$$

To arrive at this expression, we use the fact that $\left( \hat{\mathscr{P}}^{-1} \right)^{abcd} \hat{\mathscr{P}}_{cdef} = \delta^{ab}_{ef}$. We see that the new graviton mode has also been decoupled with this field redefinition and can be integrated out, leaving us with

$$\sum_{\ell m} S_{\ell m} = -\frac{1}{2} \sum_{\ell m} \int \frac{\mathrm{d}^2 p}{(2\pi)^2} \, \phi_{\ell,m} \left( p^2 + \mu^2 \lambda \right) \phi_{\ell,m}$$

$$- \sum_{\ell m} \frac{\gamma^2}{4} \int \frac{\mathrm{d}^2 p}{(2\pi)^2} \left[ \tilde{T}^{\ell m}_{ab}(-p) \hat{\mathscr{P}}^{abcd}(p) \tilde{T}^{\ell m}_{cd}(p) \right] . \tag{4.16}$$

Now, using the form of the stress tensor in (3.12), we write

$$\frac{-\gamma^2}{4} \sum_{\ell m} \int \frac{\mathrm{d}^2 p}{(2\pi)^2} \, \tilde{T}^{\ell m}_{ab}(-p) \hat{\mathscr{P}}^{abcd}(p) \tilde{T}^{\ell m}_{cd}(p) =$$

$$= \frac{-\gamma^2}{4} \sum_{\substack{\ell_1 m_1 \, \ell_3 m_3 \\ \ell_2 m_2 \, \ell_4 m_4}} \int \frac{\mathrm{d}^2 p}{(2\pi)^2} \int \left( \prod_{i=1}^4 \frac{\mathrm{d}^2 p_i}{(2\pi)^2} \, \phi_{\ell_i m_i}(p_i) \right) p_{1,a} p_{2,b} \hat{\mathscr{P}}^{abcd}(p) p_{3,c} p_{4,d}$$

$$\times C[1;2;3;4] (2\pi)^2 \delta^{(2)}(p_1 + p_2 - p)(2\pi)^2 \delta^{(2)}(p_3 + p_4 + p)$$

$$= \frac{-\gamma^2}{4} \sum_{\substack{\ell_1 m_1 \, \ell_3 m_3 \\ \ell_2 m_2 \, \ell_4 m_4}} \int \left( \prod_{i=1}^4 \frac{\mathrm{d}^2 p_i}{(2\pi)^2} \, \phi_{\ell_i m_i}(p_i) \right) p_{1,a} p_{2,b} \hat{\mathscr{P}}^{abcd}(p_1 + p_2) p_{3,c} p_{4,d}$$

$$\times C[1;2;3;4] (2\pi)^2 \delta^{(2)} \left( \sum_{i=1}^4 p_i \right) , \tag{4.17}$$

where we defined

$$C[i;j;k;l] := \sum_{\ell m} C[\ell m; \ell_i m_i; \ell_j m_j] C[\ell m; \ell_k m_k; \ell_l m_l] . \tag{4.18}$$

Therefore, the effective matter action (4.16) can be written as

$$\sum_{\ell m} S_{\ell m} = -\frac{1}{2} \sum_{\ell m} \int \frac{\mathrm{d}^2 p}{(2\pi)^2} \ \phi_{\ell,m} \left(p^2 + \mu^2 \lambda\right) \phi_{\ell,m}$$

$$-\frac{1}{4!} \sum_{\substack{\ell_1 m_1 \ \ell_3 m_3 \\ \ell_2 m_2 \ \ell_4 m_4}} \int \left(\prod_{i=1}^4 \frac{\mathrm{d}^2 p_i}{(2\pi)^2} \ \phi_{\ell_i m_i}(p_i)\right) \ (2\pi)^2 \delta^{(2)} \left(\sum_{i=1}^4 p_i\right)$$

$$3C\left[i;j;k;l\right] V\left[p_i;p_j;p_k;p_l\right] . \tag{4.19}$$

with

$$V\left[p_1;p_2;p_3;p_4\right] \ := \ 2\gamma^2 p_{1,a} p_{2,b} \hat{\mathscr{P}}^{abcd} \left(p_1 + p_2\right) p_{3,c} p_{4,d} . \tag{4.20}$$

Since the interaction term is symmetric under exchange of the different scalar partial wave legs, the corresponding Feynman rule for the vertex reads

$$i\left(C\left[1;2;3;4\right] V\left[p_1;p_2;p_3;p_4\right] + C\left[1;3;2;4\right] V\left[p_1;p_3;p_2;p_4\right] + C\left[1;4;2;3\right] V\left[p_1;p_4;p_2;p_3\right]\right) . \tag{4.21}$$

From the perspective of the original theory with the graviton, the three terms correspond to $s$, $t$, and $u$ channel scattering.

## 4.1 Approximate spherical symmetry

As shown in [4, 5], an important simplification of the theory can be obtained by fixing one of the scalars in every pair to be in the s-wave such that the Clebsch-Gordon coefficients diagonalise as

$$C\left[\ell m; \ell_i m_i; \ell_j m_j\right] \ = \ 2\delta_{\ell \ell_i} \delta_{m m_i} \quad \text{with} \quad \ell_j \ = \ 0 \ = \ m_j , \tag{4.22}$$

where the factor of two accounts for the fact that either of the scalar legs may be put in the s-wave. For the same reason, the $u$-channel (where $p_1$ and $p_3$ carry angular momentum) is projected out in this approximation. This results in the following vertex

$$2i\gamma^2 \left(p_{1,a} p_{2,b} \hat{\mathscr{P}}^{abcd} \left(p_1 + p_2\right) p_{3,c} p_{4,d} + p_{1,a} p_{4,b} \hat{\mathscr{P}}^{abcd} \left(p_1 + p_4\right) p_{2,c} p_{3,d}\right) . \tag{4.23}$$

The first term corresponds to an exchange of the transverse scalar $\mathscr{K}$ which is subleading, while the second term arises from an exchange of the longitudinal field $\mathfrak{h}_{ab}$. This result generalises the one of [4] where some further simplifying assumptions were made (for instance, external scalars were assumed to be null in that reference, while they have a small effective mass in the two-dimensional theory).

## 4.2  Scattering of s-waves

Instead of considering the approximately spherically symmetric case, another interesting possibility is to consider pure s-wave scattering of all external legs [5]. Plugging in the propagator for the graviton monopole (2.37) and taking the additional symmetry factors of identical external legs into account, the vertex now reduces to

$$\frac{\gamma^2}{\mu^2}\left[(p_1 \cdot p_2)(p_3 \cdot p_4) + (p_1 \cdot p_3)(p_2 \cdot p_4) + (p_1 \cdot p_4)(p_2 \cdot p_3)\right]. \tag{4.24}$$

**A simple proposal for the inelastic ladder of ladders**   It was shown in [5] that a certain class of infinitely many loop corrections to the tree-level $2 \to 2N$ amplitudes can be computed explicitly. These were called the 'ladder of ladders' diagrams and were computed essentially by replacing the exchanged virtual gravitons by a corresponding quantity that captures the $2 \to 2$ ladder. Several such ladders were then glued together to produce the $2 \to 2N$ ladder of ladders.

In this and the previous subsections, we saw that tree-level $2 \to 2$ amplitude is equivalent to the four-vertex interaction in the effective scalar theory. This implies that this four-vertex can be promoted to a physically relevant four-point function of interest. Therefore, replacing this tree-level vertex by the corresponding ladder and gluing several such ladders together, we immediately find a simple way to reproduce the $2 \to 2N$ ladder of ladders amplitude.

## 5  Computationally effective theory

In the previous section, we found that certain modes in the graviton decouple from cubic interactions and therefore do not contribute to the ensuing physical S-matrix. This allowed us to combine the interacting modes and the external scalar modes into an effective scalar theory with a four-vertex that captures the $2 \to 2$ graviton exchange. Pictorially, this is illustrated in Fig. 1. While this may seem to be an unnecessary rewriting of the original theory at hand, we will now argue that it is particularly efficient for computations.

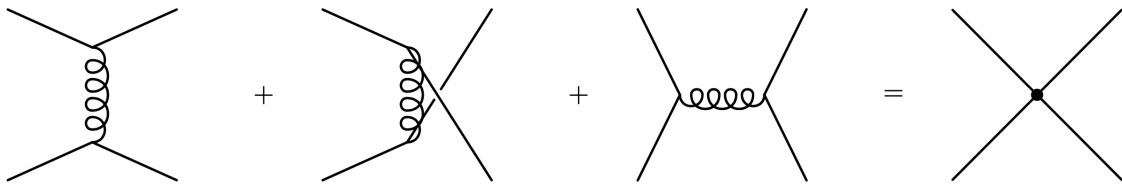

**Figure 1**. The three channels of $2 \to 2$ tree-level graviton exchange diagrams can be rewritten as a four-vertex in an effective scalar theory. This vertex is given in (4.21). As argued in Section 4.2, this vertex can be promoted to capture the $2 \to 2$ eikonal ladder of [3, 4] to capture the inelastic ladder of ladders amplitudes of [5].

One of the important utilities of this rewriting is the drastic reduction in the number of diagrams that need to be computed using the four-vertex to capture a significantly larger number of graviton exchange diagrams. This is illustrated in Fig. 2, where we consider tree-level $2 \to 4$ scattering. In the theory with graviton exchanges, there are eighteen such diagrams that contribute for fixed external legs, nine of which we draw. Rewritten in terms of the four-vertex, all of these nine diagrams are contained in the single tree-level diagram show on the right in Fig. 2. This can easily be computed and contains two copies of the four-vertex (4.21) and the scalar propagator from the quadratic term in (4.19). Therefore, the complete list of eighteen diagrams is computed by two topologically distinct scalar four-vertex diagrams.

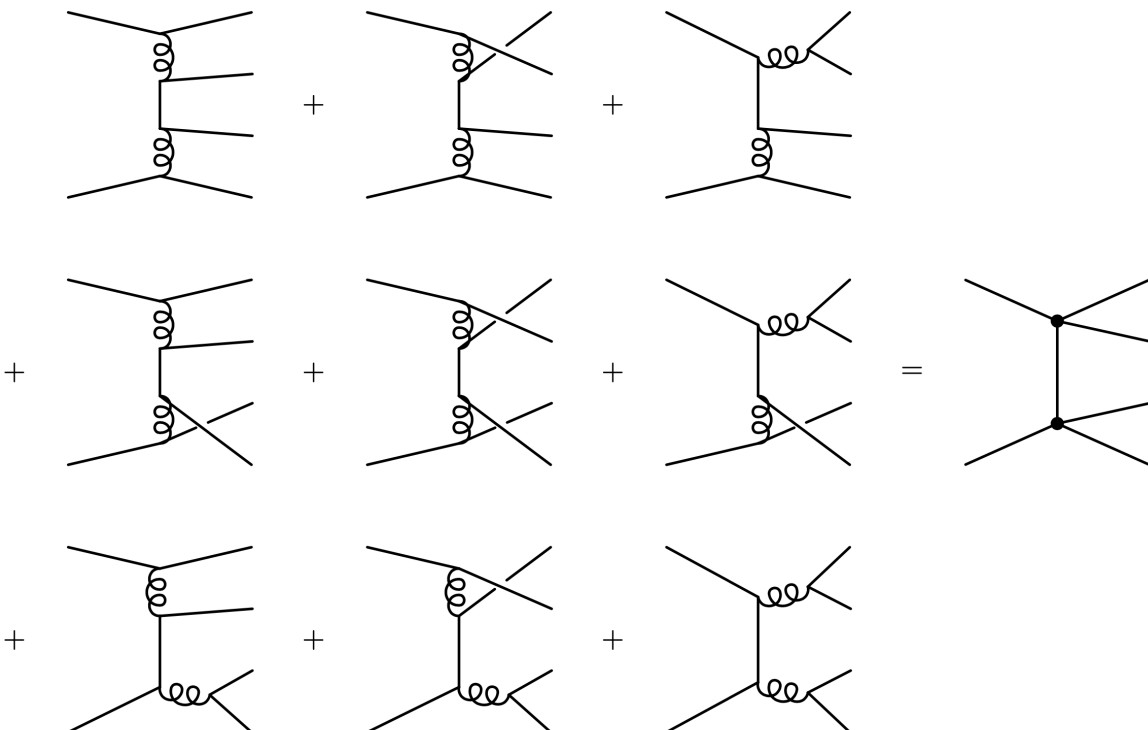

**Figure 2**. Consider nine different diagrams that capture $2 \to 4$ scattering mediated by graviton exchanges at tree-level shown on the left hand side in this figure. All of these are computed in one go by the tree-level amplitude mediated by the four-vertex (4.21) depicted pictorially on the right hand side above.

Finally, it is of particular importance to note that the utility of the four-vertex rewriting is not restricted to tree-level diagrams. It was noticed in [5] that general inelastic loop diagrams are difficult to compute. Such general diagrams were called the cobweb diagrams. As an example, consider a certain three-loop $2 \to 4$ cobweb diagram[6] shown in Fig. 3. There are

---

[6]An appropriate resummation of such diagrams is of particular interest to explore the potential existence of new chaos exponents in higher moments in the spirit of [32].

972 such diagrams, one of which we draw as a representative on the left hand side of the said figure. Such diagrams are all computed by four topologically distinct scalar three-loop diagrams of the kind shown on the right hand side of the same figure. Each such scalar diagram contains five four-vertices and each such vertex is a combination of three $2 \to 2$ diagrams as shown in Fig. 1. Therefore, the total number of diagrams contained in the four topologically distinct scalar diagrams is $4 \times 3^5 = 972$. As in Fig. 2 several diagrams of the kind on the left are captured by a single diagram of the kind on the right. At loop level, not only are the number of diagrams to be computed reduced but also the topologies of the contributing diagrams in the four-vertex theory are conceivably simpler and more tractable.

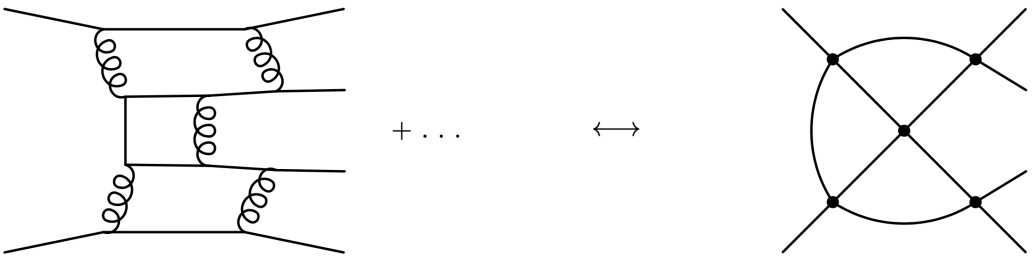

**Figure 3**. A representative of the three-loop cobweb diagrams is show on the left. Several such diagrams are captured by a single three-loop scalar diagram mediated by the four-vertex (4.21) shown on the right.

# 6 Black hole entropy from multiplicity

How black hole entropy may emerge from the scattering approach to black hole dynamics is rather mysterious. The only game in town appears to be to impose a short-distance cut-off that results in the desired result [10, 13]. Here, we make an intriguing observation that appears to emerge from the second quantised approach of the present article. The entropy contained in the multiplicity of the external particles in inelastic scattering yields almost exactly the black hole entropy.

In [5], inelastic amplitudes with particle production were computed. In particular, it was shown that in $2 \to 2N$ tree-level amplitudes grow exponentially. As mentioned in the introduction, the characteristic time delay associated with this scattering process is Page time, suggesting a breakdown of Hawking's free field theory thereafter. Of the $2N$ external particles, two sets of $N$ particles are identical. Therefore, the total multiplicity of the external states is given by

$$\Omega \ = \ \frac{(2N)!}{(N!)^2} \ \sim \ 4^N \,. \tag{6.1}$$

The corresponding entropy is then

$$S_\Omega \ \sim \ N \log(4) \,. \tag{6.2}$$

It was also shown in [5] that the $2 \to 2N$ amplitude is sharply peaked about a specific value determined by the centre of mass energy, $E$, of the scattering process

$$N_{max} = \frac{E^2 \kappa^2}{e}, \tag{6.3}$$

where $\kappa^2 = 8\pi G$. To find the entropy associated with the multiplicity of the external states, we need knowledge of the relevant centre of mass energy of the scattering process to be considered. Lacking a first principle derivation, there are several arguments to be made in favour of the choice $E = M_{BH}$:

- The canonical energy scale of the system we would like to probe is indeed $M_{BH}$.

- The Eisenbud-Wigner time-delay associated with elastic $2 \to 2$ scattering has been shown to be scrambling time, agreeing with the expectation from [11], only if we set $E = M_{BH}$ [5]. This is perhaps the strongest indication in favour of this choice.

- In similar vein, the Eisenbud-Wigner time-delay associated with inelastic $2 \to 2N$ scattering has been shown to be Page time when $E = M_{BH}$ [5]. An interpretation of this result is that when $E = M_{BH}$ energy is thrown into the black hole, it doubles the energy contained in the black hole. After the scattering process, momentum conservation implies that all the energy that went in has returned and therefore, the black hole has halved in size returning to its original mass. It is therefore natural that the half-life time of the black hole is how long one has to wait for information to return.

Therefore, with $E = M_{BH}$, we have

$$N_{max} = \frac{2}{e} 4\pi G M_{BH}^2 = \frac{2}{eG} \frac{4\pi R^2}{4} = \frac{2}{eG} A = \frac{2}{e} S_{BH}. \tag{6.4}$$

Inserting this into (6.2), we find

$$S_\Omega \sim S_{BH} \frac{2 \log (4)}{e}. \tag{6.5}$$

The proportionality factor is evidently of order one. With very little input, it is remarkable that such a result emerges!

## 7 Summary

In this article, we have developed a toolbox that can be used to compute scattering amplitudes in the presence of a fluctuating black hole background. In particular, we found the graviton propagator near the horizon of a Schwarzschild black hole in a partial wave basis for all angular momentum modes of either parity. We also showed that not all graviton modes are interacting at leading (three-point) order and found the propagator for the interacting modes. We then found that this theory can be rewritten (without loss of generality) as a

scalar theory with a particular four-vertex that captures the elastic $2 \to 2$ graviton exchange. We demonstrated that this rewriting is exceptionally effective for computations, dramatically reducing the number of computations in comparison to the original formulation.

This toolbox can be used for many interesting computations including all scattering amplitudes of interest. In addition to addressing some of the shortcomings mentioned in the introduction, it would also be interesting to address the issue of antipodal identification that appears to be a natural boundary condition to glue the future of the past horizon and the past of the future horizon together, for consistency [14, 33, 34, 35, 36, 37]. Extensions to incorporate other standard model fields and charged particles is another straightforward application. In upcoming work, we hope to address several such applications that rely on the the tools developed in this article.

## Acknowledgements

We are grateful to Gerard 't Hooft for various helpful discussions on related topics. We also thank Fabiano Feleppa and Guillermo Cortina for ongoing collaborations on related ideas. This work is supported by the Delta-Institute for Theoretical Physics (D-ITP) that is funded by the Dutch Ministry of Education, Culture and Science (OCW).

## A    Background metric and choice of coordinates

In this paper, we work in Kruskal-Szekeres coordinates that are defined as

$$xy = 2R^2 \left(1 - \frac{r}{R}\right) e^{\frac{r}{R} - 1}, \tag{A.1}$$

$$x/y = e^{2\tau}, \tag{A.2}$$

where $r(x, y)$ is implicitly defined, $\tau = \frac{t}{2R}$ and $R = 2GM$ is the Schwarzschild radius (whose inverse we call $\mu = 1/R$). We will often write these coordinates as a two-vector $x^a = \{x, y\}$ with small Latin letters denoting the longitudinal coordinates and capital Latin letters denoting the transverse coordinates on the sphere. We will also work in natural units $\hbar = c = 1$. The Schwarzschild metric in these coordinates is given by

$$ds^2 = -2A(r)dxdy + r^2(x, y) d\Omega_2^2, \tag{A.3}$$

$$A(r) = \frac{R}{r} e^{1 - \frac{r}{R}}. \tag{A.4}$$

where we employ the mostly plus signature. To leading order in the near horizon region, we have $r \sim R$ and therefore $A \sim 1$ resulting in the product space

$$ds^2 = -2dxdy + R^2 d\Omega_2^2. \tag{A.5}$$

**Christoffel symbols:** The non-vanishing Christoffel symbols of the Schwarzschild metric in Kruskal-Szekeres coordinates are given by

$$\Gamma^c_{ab} = \delta^c_{(a}U_{b)} - \frac{1}{2}g_{ab}U^c \qquad\qquad \Gamma^C_{AB} = \delta^C_{(A}W_{B)} - g_{AB}W^C \qquad (A.6)$$

$$\Gamma^A_{Ba} = V_a\delta^A_B \qquad\qquad \Gamma^a_{AB} = -V^a g_{AB}\,, \qquad (A.7)$$

where $V_a = \partial_a \log r$, $U_a = \partial_a \log A$ and $W_A = \partial_A \log \sin\theta$. The rest of the symbols $\Gamma^A_{ab} = 0 = \Gamma^b_{aA}$ vanish identically.

**The Riemann tensor:** Using the definition

$$R^\rho{}_{\mu\sigma\nu} = \partial_\sigma\Gamma^\rho_{\mu\nu} - \partial_\nu\Gamma^\rho_{\mu\sigma} + \Gamma^\rho_{\sigma\kappa}\Gamma^\kappa_{\mu\nu} - \Gamma^\rho_{\mu\kappa}\Gamma^\kappa_{\sigma\nu} \qquad (A.8)$$

the only non-vanishing components of the Riemann tensor are related to the following components by symmetries

$$R_{xyxy} = \partial_x\partial_y \log A\,, \qquad (A.9)$$

$$R_{\theta\phi\theta\phi} = r^2\sin^2\theta\left(1 + \frac{2\partial_x r \partial_y r}{A}\right)\,, \qquad (A.10)$$

$$R_{aAbB} = g_{AB}S_{ab}\,, \qquad (A.11)$$

where in the last line we defined a new tensor

$$S_{ab} := -\nabla_{(a}V_{b)} - V_a V_b = -\tilde{\nabla}_{(a}V_{b)} - V_a V_b\,. \qquad (A.12)$$

In the second equality above, we defined $\tilde{\nabla}$ which is the covariant derivative only along the longitudinal directions. This second equality holds because $\Gamma^C_{ab} = 0$.

## A.1 The antisymmetric Levi-Civita tensor

Here we define the antisymmetric tensor on the transverse two-sphere as

$$\epsilon_{AB} = r^2\sin\theta\begin{pmatrix} 0 & 1 \\ -1 & 0 \end{pmatrix}\,, \qquad (A.13)$$

with $\epsilon_{\theta\phi} = r^2\sin\theta = -\epsilon_{\phi\theta}$. Indices are raised and lowered with the usual round metric on the two-sphere. Therefore we have

$$\epsilon_A{}^B = \begin{pmatrix} 0 & \sin\theta \\ -\csc\theta & 0 \end{pmatrix} \quad\text{and}\quad \epsilon^{AB} = \frac{1}{r^2\sin\theta}\begin{pmatrix} 0 & 1 \\ -1 & 0 \end{pmatrix}\,. \qquad (A.14)$$

Similarly, in the near horizon region $r \sim R$, we have the antisymmetric tensor in the longitudinal directions

$$\epsilon^{ab} = \begin{pmatrix} 0 & 1 \\ -1 & 0 \end{pmatrix} \quad\text{and}\quad \epsilon_{ab} = \begin{pmatrix} 0 & 1 \\ -1 & 0 \end{pmatrix}\,. \qquad (A.15)$$

## B    The group of traceless tensor operators

In this section, we consider the group $\bar{G}$ consisting of all invertible rank four tensors that are (symmetric) traceless in their first and last pair of indices. For $M, N \in \bar{G}$ the elements in the group are defined by

$$\eta^{ab} M_{abcd} \;=\; 0 \;=\; M_{abcd} \eta^{cd} \,. \tag{B.1}$$

The group operation is given by

$$(M \cdot N)_{abef} \;\equiv\; M_{abcd} N^{cd}{}_{ef} \,, \tag{B.2}$$

where closure $(M \cdot N \in \bar{G})$ is guaranteed by the observation that $M \cdot N$ is still traceless over $ab$ and $ef$. Of course, $\bar{G}$ is a subgroup of the group $G$ that consists of *all* invertible rank four tensors. The question of interest now is what the identity element $\bar{I} \in \bar{G}$ is, since this might differ from the identity element $I \in G$ given by $I^{cd}_{ab} = \delta^c_{(a} \delta^d_{b)}$. Indeed we require $\bar{I}$ to be traceless, but $I$ clearly isn't. The obvious correction would be to make it traceless

$$\bar{I}^{cd}_{ab} \;=\; \delta^c_{(a} \delta^d_{b)} - \frac{1}{2} \eta_{ab} \eta^{cd} \,. \tag{B.3}$$

It can now be checked that

$$\eta^{ab} \bar{I}^{cd}_{ab} = 0 = \bar{I}^{cd}_{ab} \eta_{cd} \quad \text{and} \quad \left( \bar{I} \cdot M \right)_{abef} = \bar{I}^{cd}_{ab} M_{cdef} = M_{abef} \tag{B.4}$$

hold, since $M$ is traceless. Therefore, we have

$$M^{-1} \cdot M \;=\; M^{-1}_{abcd} M^{cd}{}_{ef} \;=\; \bar{I}_{abef} \;=\; \bar{I} \,. \tag{B.5}$$

where we used that $\bar{I}_{abef} = \bar{I}_{efab}$ which ensures that there is no ambiguity between using covariant or contravariant indices.

## C    Calculation of the propagator for the odd harmonics

We begin with the first line of the action (2.1):

$$
\begin{aligned}
S &\;=\; -\frac{1}{2} \int \mathrm{d}^4 x \sqrt{-g} h^{\mu\nu} \left[ \frac{1}{2} \left( 2 \nabla^\rho \nabla_{(\mu} h_{\nu)\rho} - \Box h_{\mu\nu} - \nabla_\mu \nabla_\nu h \right) - \frac{1}{2} g_{\mu\nu} \left( \nabla^\rho \nabla^\sigma h_{\rho\sigma} - \Box h \right) \right] \\
&\;=\; -\frac{1}{2} \int \mathrm{d}^4 x \sqrt{-g} h^{\mu\nu} \left[ G^{(1)}_{\mu\nu} \right] \,,
\end{aligned}
\tag{C.1}
$$

where $G^{(1)}_{\mu\nu}$ is the first order variation of the Einstein tensor. Into this action, we will now plug in the odd harmonics

$$h^-_{aA} \;=:\; h_a \eta_A \quad \text{with} \quad \eta_A \;:=\; -\epsilon_A{}^B \partial_B Y^m_\ell \,. \tag{C.2}$$

First, we notice that $\eta_A$ does not depend on the longitudinal coordinates while $h_a$ is independent of the transverse spherical coordinates. Since $g_{aA} = 0$ and $h^- = g^{\mu\nu}h^-_{\mu\nu} = 0$, we have

$$G^{(1),-}_{aA} = \frac{1}{2}g^{\rho\sigma}\left(\nabla_\rho\nabla_a h^-_{A\sigma} + \nabla_\rho\nabla_A h^-_{a\sigma}\right) - \frac{1}{2}\Box h^-_{aA}. \tag{C.3}$$

We now use the familiar property $[\nabla_\rho, \nabla_\sigma]h_{\mu\nu} = -R^\tau{}_{\mu\rho\sigma}h_{\tau\nu} - R^\tau{}_{\nu\rho\sigma}h_{\tau\mu}$, to rewrite the above as

$$G^{(1),-}_{aA} = \nabla_{(a}L_{A)} - \frac{1}{2}g^{\rho\sigma}\left(R^\tau{}_{A\rho a}h^-_{\tau\sigma} + R^\tau{}_{\sigma\rho a}h^-_{\tau A} + R^\tau{}_{a\rho A}h^-_{\tau\sigma} + R^\tau{}_{\sigma\rho A}h^-_{\tau a}\right) - \frac{1}{2}\Box h^-_{aA}, \tag{C.4}$$

where we defined $L_\mu := \nabla^\nu h^-_{\mu\nu}$. Using the Riemann tensor components written out in Appendix A and writing out the covariant derivatives, we find

$$G^{(1),-}_{aA} = \frac{1}{2}\left(\partial_a - 2V_a\right)L_A + \frac{1}{2}\partial_A L_a + S_{ab}h^b\eta_A - \frac{1}{2}\Box h^-_{aA}, \tag{C.5}$$

where we used $R_{AaBb}h^{bB}_- = S_{ab}h^b\eta_A$ from the definition of the quantity $S_{ab}$ in Appendix A. We will now calculate the quantities $L_\mu$ and the $\Box$ term. The first quantity to calculate is

$$\begin{aligned}
L_A &= g^{\mu\nu}\nabla_\mu h^-_{A\nu} \\
&= g^{\mu\nu}\partial_\mu h^-_{A\nu} - g^{\mu\nu}\Gamma^\rho_{\mu A}h^-_{\rho\nu} - g^{\mu\nu}\Gamma^\rho_{\mu\nu}h^-_{A\rho} \\
&= \eta_A g^{ab}\left(\partial_a + 2V_a\right)h_b,
\end{aligned} \tag{C.6}$$

where we inserted $h^{aA}_- = h^a\eta^A$ and the relevant Christoffel symbols to arrive at the last line. Similarly,

$$\begin{aligned}
L_a &= g^{\mu\nu}\nabla_\mu h^-_{a\nu} \\
&= g^{\mu\nu}\partial_\mu h^-_{a\nu} - g^{\mu\nu}\Gamma^\rho_{\mu a}h^-_{\rho\nu} - g^{\mu\nu}\Gamma^\rho_{\mu\nu}h^-_{a\rho} \\
&= h_a\frac{1}{\sin\theta}g^{AB}\partial_A\left(\sin\theta\,\eta_B\right) \\
&= h_a\hat{\nabla}^A\eta_A \\
&= -h_a\hat{\nabla}^A\epsilon_{BC}\hat{\nabla}^C Y_{\ell m} \\
&= 0,
\end{aligned} \tag{C.7}$$

where in the second last line, we identified the covariant derivative on the two-sphere and to arrive at the last line, we observe that the covariant derivatives acting on $Y_{\ell m}$ are in competition with the antisymmetry of $\epsilon_{BC}$.

The term containing the d'Alembertian can be written as

$$\begin{aligned}
\Box h^-_{aA} &= \Box h^-_{aA} - \partial^c\left(\Gamma^b_{ca}h^-_{bA}\right) - \partial^\nu\left(\Gamma^B_{\nu A}h^-_{Ba}\right) - g^{\mu\nu}\Gamma^\tau_{\mu\nu}\left(\partial_\tau h^-_{aA} - \Gamma^b_{\tau a}h^-_{bA} - \Gamma^B_{\tau A}h^-_{Ba}\right) \\
&\quad - g^{\mu\nu}\Gamma^b_{\mu a}\partial_\nu h^-_{bA} + g^{\mu\nu}\Gamma^\tau_{\mu a}\Gamma^b_{\nu\tau}h^-_{bA} - g^{\mu\nu}\Gamma^B_{\mu A}\partial_\nu h^-_{Ba} \\
&\quad + g^{\mu\nu}\Gamma^\tau_{\mu A}\Gamma^B_{\nu\tau}h^-_{Ba} + 2g^{\mu\nu}\Gamma^\tau_{\mu a}\Gamma^\kappa_{\nu A}h^-_{\kappa\tau}.
\end{aligned} \tag{C.8}$$

We now insert $h_{aA}^- = h_{Aa}^- = h_a \eta_A$ and separate the longitudinal terms from the transverse ones. This gives

$$
\begin{aligned}
\Box h_{aA}^- &= \eta_A \Big[ g^{cd} \partial_c \partial_d h_a - \partial^c \left( \Gamma_{ac}^b h_b \right) - \partial^b \left( V_b h_a \right) - g^{\mu\nu} \Gamma_{\mu\nu}^b \partial_b h_a + g^{\mu\nu} \Gamma_{\mu\nu}^c \Gamma_{ac}^b h_b + g^{\mu\nu} \Gamma_{\mu\nu}^b V_b h_a \\
&\qquad - \Gamma_{ac}^b \partial^c h_b + g^{cd} \Gamma_{ca}^e \Gamma_{de}^b h_b - 2 V_a V^b h_b - V_b \partial^b h_a + 2 V^c \Gamma_{ca}^b h_b - 2 V_a V^b h_b \Big] \\
&\quad + h_a \Big[ g^{BC} \partial_B \partial_C \eta_A - \partial^C \left( \Gamma_{AC}^B \eta_B \right) - g^{\mu\nu} \Gamma_{\mu\nu}^B \partial_B \eta_A + g^{\mu\nu} \Gamma_{\mu\nu}^C \Gamma_{AC}^B \eta_B \\
&\qquad - \Gamma_{AC}^B \partial^C \eta_B + g^{CD} \Gamma_{CA}^E \Gamma_{DE}^B \eta_B \Big] .
\end{aligned}
\tag{C.9}
$$

Writing longitudinal covariant derivatives with a tilde and transverse ones with a hat, this can be compactly written as

$$
\Box h_{aA}^- = \eta_A \tilde{\Box} h_a - 4 \eta_A V_a V^b h_b - 2 \eta_A V^b V_b h_a - \eta_A \left( \partial^b V_b \right) h_a + h_a \hat{\Delta} \eta_A ,
\tag{C.10}
$$

where $\hat{\Delta}$ is the Laplacian on the two sphere.

## C.1 Integrating the two-sphere out

Piecing the above terms together, we have:

$$
\begin{aligned}
G_{aA}^{(1),-} &= \frac{1}{2} \eta_A \Big[ -\tilde{\Box} h_a + \left( \tilde{\nabla}_a - 2 V_a \right) \left( \tilde{\nabla}_b + 2 V_b \right) h^b + 2 S_{ab} h^b + 4 V_a V_b h^b + 2 V^2 h_a \\
&\qquad + \left( \tilde{\nabla} \cdot V \right) h_a \Big] + \frac{1}{2} h_a \tilde{\Box} \eta_A .
\end{aligned}
\tag{C.11}
$$

We notice that

$$
\begin{aligned}
\hat{\Box} \eta_A &= -\epsilon^{AB} \eta^{CD} \hat{\nabla}_C \hat{\nabla}_D \hat{\nabla}_B Y_{\ell m} \\
&= -\epsilon^{AB} \left( \eta^{CD} \hat{\nabla}_C \hat{\nabla}_B \hat{\nabla}_D Y_{\ell m} \right) \\
&= -\epsilon^{AB} \left( \hat{\nabla}_B \hat{\Box} Y_{\ell m} - \eta^{CD} \hat{R}_{DCB}^E \hat{\nabla}_E Y_{\ell m} \right) \\
&= -\epsilon^{AB} \left( \hat{\nabla}_B \hat{\Box} Y_{\ell m} + \hat{R}_B^E \hat{\nabla}_E Y_{\ell m} \right) \\
&= -\epsilon^{AB} \left( \hat{\nabla}_B \hat{\Box} Y_{\ell m} + \hat{R} \hat{\nabla}_B Y_{\ell m} \right) \\
&= -\frac{\lambda - 2}{r^2} \eta_A ,
\end{aligned}
\tag{C.12}
$$

where the second equality is allowed because the spherical harmonics are scalar functions on the sphere, and in the last equality, we used that the Ricci tensor of the two sphere satisfies $\hat{R}_{AB} = \frac{1}{2} \hat{R} g_{AB} = \frac{1}{r^2} g_{AB}$.

Therefore, the action for the odd harmonics can now be written as

$$
\begin{aligned}
S &= -\sum_{\ell,m,\ell',m'} \int \mathrm{d}^4 x \sqrt{-g} h^a_{\ell m} \eta^A_{\ell m} \left[ G^{(1),-}_{aA,\ell'm'} \right] \\
&= -\sum_{\ell,m,\ell',m'} \frac{1}{2} \int \mathrm{d}^2 x \; A(x,y) \, r(x,y)^2 \int \mathrm{d}\Omega_2 \; \eta^A_{\ell m} \eta_{A,\ell'm'} \\
&\qquad h^a_{\ell m} \left[ \left( -\tilde{\Box} + 2V^c V_c + \left( \tilde{\nabla} \cdot V \right) + \frac{\lambda-2}{r^2} \right) g_{ab} + \left( \tilde{\nabla}_a - 2V_a \right) \left( \tilde{\nabla}_b + 2V_b \right) \right. \\
&\qquad\qquad \left. + 2S_{ab} + 4V_a V_b \right] h^b_{\ell'm'} \\
&= -\frac{\lambda-1}{2} \sum_{\ell,m} \int \mathrm{d}^2 x \; A(x,y) \, h^a_{\ell m} \left[ \left( -\tilde{\Box} + 2V^c V_c + \left( \tilde{\nabla} \cdot V \right) + \frac{\lambda-2}{r^2} \right) g_{ab} \right. \\
&\qquad\qquad \left. + \left( \tilde{\nabla}_a - 2V_a \right) \left( \tilde{\nabla}_b + 2V_b \right) + 2S_{ab} + 4V_a V_b \right] h^b_{\ell m}, \qquad\qquad \text{(C.13)}
\end{aligned}
$$

where in the third equality, we inserted the definition of $\eta_A$ and applied Stokes' theorem before integrating over the sphere. Remarkably, the action is proportional to $\lambda - 1 = \ell^2 + \ell$. Therefore, the contribution of these odd modes for large $\ell$ is heavily suppressed in comparison to the even modes.

## C.2 Weyl transformation and the near-horizon approximation

The quadratic operator in (C.13) reads

$$
\begin{aligned}
\mathscr{D}_{ab} &= A(x,y) \left[ \left( -\tilde{\Box} + 2V^c V_c + \left( \tilde{\nabla} \cdot V \right) + \frac{\lambda-2}{r^2} \right) g_{ab} + \left( \tilde{\nabla}_a - 2V_a \right) \left( \tilde{\nabla}_b + 2V_b \right) \right. \\
&\qquad\qquad \left. + 2S_{ab} + 4V_a V_b \right] \\
&= A(x,y) \left[ \left( -\tilde{\Box} + 2V^c V_c + \left( \tilde{\nabla} \cdot V \right) + \frac{\lambda-2}{r^2} \right) g_{ab} + \tilde{\nabla}_a \tilde{\nabla}_b + 2 \left( \tilde{\nabla}_a V_b \right) + 2V_b \tilde{\nabla}_a \right. \\
&\qquad\qquad \left. - 2V_a \tilde{\nabla}_b + 2S_{ab} \right] \\
&= A(x,y) \left[ \left( -\tilde{\Box} + 2V^c V_c + \left( \tilde{\nabla} \cdot V \right) + \frac{\lambda-2}{r^2} \right) g_{ab} + \tilde{\nabla}_a \tilde{\nabla}_b + 2 \left( \tilde{\nabla}_a V_b \right) + 2V_b \tilde{\nabla}_a \right. \\
&\qquad\qquad \left. - 2V_a \tilde{\nabla}_b - 2V_a V_b \right]
\end{aligned}
$$
$$
\text{(C.14)}
$$

where in the third equality, we plugged in the definition of $S_{ab}$ from (A.12). Given the complicated background spacetime, this operator is not invertible. However, we will now exploit the conformal flat nature of the metric $g_{ab} = A(x,y) \eta_{ab}$ to make the following field redefinition

$$
h_a = \sqrt{A(x,y)} \, \mathfrak{h}_a . \qquad\qquad \text{(C.15)}
$$

This transforms the quadratic operator as

$$\mathcal{D}_{ab} \longrightarrow \sqrt{A(x,y)}\mathcal{D}_{ab}\sqrt{A(x,y)}. \tag{C.16}$$

In order to cycle the $\sqrt{A(x,y)}$ from the far right in the above equation to the other side, we will make extensive use of the following identity

$$\partial_a \sqrt{A(x,y)} = \sqrt{A(x,y)}\left(\partial_a + \frac{1}{2}U_a\right). \tag{C.17}$$

We will now write all quantities appearing in the odd harmonic action (C.13) in terms of the Weyl transformed field and the flat metric $\eta_{ab}$. Therefore, in what follows, we will use the symbol "$\rightarrow$" to show the step where the Weyl transformation and replacement of $g_{ab}$ by $A(x,y)\eta_{ab}$ have been made.

The first term of interest is

$$
\begin{aligned}
\tilde{\nabla}^a \tilde{\nabla}^b h_b &= \partial^a \partial^b h_b \\
&\rightarrow A\sqrt{A}\frac{1}{A}\partial^a \frac{1}{A}\partial^b \sqrt{A}\, \mathfrak{h}_b \\
&= \left(\partial^a - \frac{1}{2}U^a\right)\left(\partial^b + \frac{1}{2}U^b\right)\mathfrak{h}_b. 
\end{aligned} \tag{C.18}
$$

As desired there are only flat space derivatives and potential terms that are artefacts of the curvature. Next, we consider

$$
\begin{aligned}
\tilde{\nabla}_a \tilde{\nabla}^a h_b &= g^{ac}\tilde{\nabla}_a \tilde{\nabla}_c h_b \\
&= g^{cd}\partial_c \partial_d h_b - U_c \partial^c h_b - U_b \partial^c h_c + U^a \partial_b h_a - \frac{1}{2}(\partial^c U_c)h_b - \frac{1}{2}(\partial^c U_b)h_c + \frac{1}{2}(\partial_b U_c)h^c \\
&\rightarrow g^{cd}\left(\partial_c + \frac{1}{2}U_c\right)\left(\partial_d + \frac{1}{2}U_d\right)\mathfrak{h}_b - U_c\left(\partial^c + \frac{1}{2}U^c\right)\mathfrak{h}_b - U_b\left(\partial^c + \frac{1}{2}U^c\right)\mathfrak{h}_c \\
&\quad + U^c\left(\partial_b + \frac{1}{2}U_b\right)\mathfrak{h}_c - \frac{1}{2}(\partial^c U_c)\mathfrak{h}_b - \frac{1}{2}(\partial^c U_b)\mathfrak{h}_c + \frac{1}{2}(\partial_b U_c)\mathfrak{h}^c \\
&= \eta_{bc}\left[\partial^2 - \frac{1}{4}U_d U^d\right]\mathfrak{h}^c + \left[-U_b \partial_c + U_c \partial_b\right]\mathfrak{h}^c, 
\end{aligned} \tag{C.19}
$$

where in the second equality, we used that $g^{cd}\Gamma^e_{cb}\Gamma^a_{de} = 0$. Finally we consider the single derivative terms:

$$
\begin{aligned}
\tilde{\nabla}^a h_b &= \partial^a h_b - \frac{1}{2}U_b h^a - \frac{1}{2}U^a h_b + \frac{1}{2}\delta^a_b U^c h_c \\
&\rightarrow \partial^a \mathfrak{h}_b - \frac{1}{2}U_b \mathfrak{h}^a + \frac{1}{2}\delta^a_b U^c \mathfrak{h}_c. 
\end{aligned} \tag{C.20}
$$

Putting these results together gives the following operator:

$$
\begin{aligned}
\mathcal{D}_{ab} &\rightarrow \partial_a \partial_b + U_{[a}\partial_{b]} - 4V_{[a}\partial_{b]} + \frac{1}{2}(\partial_a U_b) - \frac{1}{4}U_a U_b - 2V_a V_b \\
&\quad + \eta_{ab}\left(-\partial^2 + \frac{1}{4}U_d U^d - V_c U^c + 2V_c V^c + (\partial_c V^c) + A(x,y)\frac{\lambda - 2}{r^2}\right). 
\end{aligned} \tag{C.21}
$$

The biggest advantage of all the manipulations done so far is that the theory is now entirely defined in two dimensional flat space with the Minkowski metric $\eta_{ab}$. The remaining derivatives are all partial derivatives with the curvature traded for complicated potential terms. This allows us the luxury of defining familiar a Fourier transforms. So far all calculations were exact. The problem still remains that the operator is not invertible analytically. To remedy this problem, we will employ a near-horizon approximation to find an inverse near the horizon.

### The near-horizon approximation

To leading order, near the horizon we have $r \sim R$. This implies that the longitudinal coordinates satisfy $x, y \ll R$ or equivalently $\mu x_a \ll 1$. The potentials now become

$$V_a \sim \frac{1}{2}\mu^2 x_a \quad \text{and} \quad U_b \sim -\mu^2 x_b. \tag{C.22}$$

The quadratic operator therefore simplifies to

$$\mathscr{D}^{ab} = \partial^a \partial^b - \eta^{ab}\partial^2 - 3\mu^2 x^{[a}\partial^{b]} + \mu^2\eta^{ab}\left(\lambda - \frac{3}{2}\right). \tag{C.23}$$

Therefore, the action for the odd modes can be written as

$$\sum_{\ell,m} S_{\ell,m} = \sum_{\ell,m} -\frac{\lambda-1}{2} \int \mathrm{d}^2x \, \mathfrak{h}^a \left(\partial_a\partial_b - \eta_{ab}\partial^2 - 3\mu^2 x_{[a}\partial_{b]} + \mu^2\eta_{ab}\left(\lambda - \frac{3}{2}\right)\right)\mathfrak{h}^b. \tag{C.24}$$

The quadratic operator is now easy to invert, using ideas developed in [4].

### C.3  Propagator for the odd harmonics

To invert the operator, we first perform a Fourier transform, resulting in

$$\mathscr{D}^{ab} = \eta^{ab}\left(k^2 + \mu^2\left(\lambda - \frac{3}{2}\right)\right) - k^a k^b - 3\mu^2 k^{[a}\partial_k^{b]}, \tag{C.25}$$

where we used that

$$
\begin{aligned}
\int \mathrm{d}^2x\mathfrak{h}^a\left(-3\mu^2 x_{[a}\partial_{b]}\right)\mathfrak{h}^b &= \int \mathrm{d}^2x \int \mathrm{d}^2k \int \mathrm{d}^2k'\mathfrak{h}^a(k)\,\mathfrak{h}^b(k')\,e^{ik\cdot x}\left(-3\mu^2 x_{[a}\partial_{b]}\right)e^{ik'\cdot x} \\
&= \int \mathrm{d}^2x \int \mathrm{d}^2k \int \mathrm{d}^2k'\mathfrak{h}^a(k)\,\mathfrak{h}^b(k')\,e^{ik\cdot x}\left(-3i\mu^2 x_{[a}k'_{b]}\right)e^{ik'\cdot x} \\
&= \int \mathrm{d}^2x \int \mathrm{d}^2k \int \mathrm{d}^2k'\mathfrak{h}^a(k)\,\mathfrak{h}^b(k')\,e^{ik\cdot x}\left(-3\mu^2 k'_{[b}\partial_{a]}^{k'}\right)e^{ik'\cdot x} \\
&= \int \mathrm{d}^2x \int \mathrm{d}^2k \int \mathrm{d}^2k'e^{i(k+k')\cdot x}\mathfrak{h}^a(k)\left(3\mu^2 \partial_{[a}^{k'} k'_{b]}\right)\mathfrak{h}^b(k') \\
&= (2\pi)^2 \int \mathrm{d}^2k'\mathfrak{h}^a(-k')\left(-3\mu^2 k'_{[a}\partial_{b]}^{k'}\right)\mathfrak{h}^b(k'), \tag{C.26}
\end{aligned}
$$

where in the second equality we rewrote the partial derivative, in the third, we commuted $x$ and $k$ before rewriting $x$ as a partial derivative with respect to $k$, integrated by parts in the fourth, and used antisymmetry in the final equality.

We would now like to find the Green's function $\mathcal{G}^{bc}$ that satisfies

$$\mathcal{D}_{ab}\, \mathcal{G}^{bc} \;=\; \delta_a^c \,. \tag{C.27}$$

Dilation invariance of the background $x \to \lambda x$, $y \to \lambda^{-1} y$ corresponds to time translation invariance of the black hole background. This suggests the following form for the Green's function

$$\mathcal{G}^{bc} \;=\; F\left(k^2\right)\left(\eta^{bc} + G\left(k^2\right) k^b k^c\right), \tag{C.28}$$

where $F$, $G$ are to be determined. Acting with the operator $\mathcal{D}$ on $\mathcal{G}$ gives

$$\mathcal{D}_{ab}\, \mathcal{G}^{bc} \;=\; F\delta_a^c \left(k^2 + \mu^2\left(\lambda - \frac{3}{2}\right)\right) + F k_a k^c \left(G\mu^2\left(\lambda - \frac{3}{2}\right) - 1\right)$$
$$- 3\mu^2\left(k_{[a}\partial_k^{c]} F + k_{[a}\partial_{b]}^k\left(FGk^b k^c\right)\right). \tag{C.29}$$

Action of the the derivatives on $F$, $G$ can be worked out using the chain rule resulting in $\partial_a F = 2F' k_a$ and $\partial_a G = 2G' k_a$. However, since all derivatives contain an antisymmetrisation, we see that derivatives on the scalar factors vanish:

$$k_{[a}\partial_k^{c]} F \;=\; 2F' k_{[a} k^{c]} \;=\; 0\,. \tag{C.30}$$

Therefore, the total contribution from the derivatives is given by

$$k_{[a}\partial_k^{c]} F + k_{[a}\partial_{b]}^k\left(k^b k^c F G\right) \;=\; FG k_{[a}\delta_{b]}^b k^c + FG k_{[a}\delta_{b]}^c k^b$$
$$= \frac{1}{2}FG\left(k_a k^c \delta_b^b - k_b k^c \delta_a^b + k_a k^b \delta_b^c - k^2 \delta_a^c\right)$$
$$= FG k_a k^c - \frac{1}{2}FG k^2 \delta_a^c\,. \tag{C.31}$$

Inserting this contribution into the defining equation gives

$$\mathcal{D}_{ab}\, \mathcal{G}^{bc} \;=\; F\delta_a^c\left(k^2 + \mu^2\left(\lambda - \frac{3}{2}\right) + \frac{3}{2}\mu^2 G k^2\right) + F k_a k^c\left(G\mu^2\left(\lambda - \frac{9}{2}\right) - 1\right) \;=\; \delta_a^c\,. \tag{C.32}$$

The solution to this equation is easily seen to be given by

$$F \;=\; \frac{\lambda - \frac{9}{2}}{\lambda - 3}\frac{1}{k^2 + \mu^2 \frac{\left(\lambda - \frac{3}{2}\right)\left(\lambda - \frac{9}{2}\right)}{\lambda - 3}} \quad \text{and} \quad G \;=\; \frac{1}{\mu^2\left(\lambda - \frac{9}{2}\right)}\,. \tag{C.33}$$

We now finally have the propagator for the odd harmonics which reads

$$\mathcal{G}^{ab} \;=\; \frac{\lambda - \frac{9}{2}}{\lambda - 1}\frac{1}{k^2\left(\lambda - 3\right) + \mu^2\left(\lambda - \frac{3}{2}\right)\left(\lambda - \frac{9}{2}\right)}\left(\eta^{ab} + \frac{k^a k^b}{\mu^2\left(\lambda - \frac{9}{2}\right)}\right). \tag{C.34}$$

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
