# Peer review of "A toolbox for black hole scattering"

_SciPost Physics_

## Round 1 · Referee Report · Anonymous (Referee 1) · 2023-11-6

Report

This paper aims to construct a framework for investigating particle scattering off a black hole.

The introduction is both comprehensive and compelling in its motivation of the proposed method. The strengths and weaknesses of the method are articulated with clarity, and the presentation of calculations is both lucid and succinct.

However, my principal concern regarding the paper is the absence of concrete applications for their method. While the paper alludes to a reduction in the number of diagrams, it remains unclear what insights can be gleaned from this approach. As hinted at in the introduction, this method holds the potential to study Hawking radiation, which is a highly interesting prospect, yet it still appears somewhat out of reach. Therefore, I strongly recommend that the authors elaborate on the applications of their current method, either within the introduction or the summary section.

Additionally, I kindly request the authors to provide a clear definition of odd and even graviton modes in the preamble of section 2. Although a reference has been cited, this is essential for completeness.

In light of the aforementioned concerns and recommendations for improvement, I recommend this paper for publication after minor revisions.

---

## Round 1 · Referee Report · Anonymous (Referee 2) · 2023-12-11

Strengths

1) clear presentation of a concrete and relevant calculation

2) discussion of an interesting reformulation of scalar gravitational three-point interactions as an effective scalar four-point interaction

3) detail with regards to different gauge/coordinate choices in the fluctuations around Schwarzschild

Weaknesses

1) relation/comparison to previous work and, in particular, related approaches

2) unclear distinction between nonlinear classical corrections and quantum corrections

3) somewhat vague and potentially confusing discussion of potential phenomenological importance

4) incomplete discussion of the limitations of their toolbox/approach

Report

Completing their previous work, the authors carefully detail a calculation of the graviton propagator and the graviton-scalar-scalar three-point function in the near-horizon region of a Schwarzschild background spacetime. In a formulation similar to that of the Regge-Wheeler-Zerilli ansatz to black-hole perturbation theory, they provide great detail on different gauge choices and give a partial comparison to existing literature on the topic. Moreover, they find a way to rewrite the 2-to-2 scalar scattering, arising from the above truncation, as an effective four-point vertex and demonstrate that this reformulation can be used to simplify calculational effort. These results are presented with great care and detail and are of importance to the treatment of quantized gravitational interactions in Schwarzschild spacetime and, hence, warrant publication in SciPost.

In addition, the authors discuss relations to (i) Hawking radiation and scrambling/Page time (in Sec 1), (ii) to a definition of black-hole entropy (in Sec 6), and potential phenomenology in post-merger gravitational-wave signals (in Sec 1). In my view, these discussions remain rather incomplete and unconvincing for the reasons detailed below.
As I also detail below, in my opinion, the reader would benefit from a (even more) detailed discussion of gauge dependence and a comparison to existing literature.

Requested changes

Related to the above, I ask the authors for clarification and improvement of the manuscript regarding my following major comments (1-4):

1) The question of gauge dependence and dynamical degrees of freedom seems to remain somewhat unresolved. The reader would benefit from a more conclusive discussion. Similarly, the authors should emphasize that their results are limited to Schwarzschild spacetime and might want to comment on a potential generalization to Kerr (or even dynamical) spacetimes.

2) It seems to me that the presented calculation of the graviton propagator and the graviton-scalar-scalar vertex -- at least in intermediate steps -- should also agree (in appropriate limits) with other available literature, in particular, on black-hole perturbation theory (see, e.g., https://doi.org/10.1088/0264-9381/16/12/201 and related references), scattering amplitudes (see, e.g., https://arxiv.org/abs/2003.08351 and related references), the effective field theory approach (see, e.g., https://arxiv.org/pdf/hep-th/0211071.pdf and related references) and functional approaches (see, e.g., https://arxiv.org/pdf/2204.08564.pdf and related references). In an attempt to crosscheck and unify calculations, the reader would greatly benefit from a concrete comparison with this previous literature.

3) On p.4, the authors provide a paragraph on "observational consequences" which remains somewhat confusing to me. Most importantly, they write: "Finally, in analogy to the Post-Newtonian and Post-Minkowskian expansions associated with the inspiral phase, a natural Post-BH expansion possibly governs the post-merger phase that improves upon the classical ringdown prediction." The PN/PM expansion is an expansion of classical GR, hence, resumms to full nonlinear but still classical GR, and does not account for quantization. Modern amplitude calculations contain and can reproduce these classical pieces but also contain genuine quantum effects (see, for instance, https://arxiv.org/abs/hep-th/0405239v1). Previous statements of the authors about gravitational-wave echos and their stated expectation "that black hole scattering will contribute to the gravitational wave signals of the post-merger phase of the compact binary mergers observed in nature" seem to further conflate nonlinear classical terms with quantum corrections. The reader would greatly benefit from a more elucidating discussion of how the authors' work related to the above. In particular: Do the authors expect that the quantum (not the nonlinear classical) corrections contained in their calculation are observationally relevant for astrophysical black holes?

4) The authors seem to put great emphasis on the fact that their work presents a generic toolbox, however, they do not comment much on limitations. I would expect the following extensions to be very nontrivial: (i) extensions which include other vertices such as the scalar-scalar-graviton-graviton vertex; (ii) extensions to other matter fields; (iii) extensions to non-minimal couplings, (iv) extensions beyond the near-horizon regime. I understand that all of this is beyond the scope of the present work, however, at times, the authors make it seem as if the main step to all such extensions has been taken and all of this follows as a mere application of their results. The reader would benefit greatly from a more concrete discussion.

Once the above points are suitably addressed, I can confidently recommend the paper for publication in SciPost.

In addition, I have the following minor remarks, which the authors may or may not want to consider:

a) While not central to the impact of their calculation, the discussion of entropy remains unconvincing to me. In particular, I cannot see why one should identify the centre-of-mass energy E of the scattering process with the black-hole mass or, vice versa, why the entropy should follow from this specific choice of scattering process.

b) The reader would benefit from suitable citations regarding the scrambling time and the Page time.

c) The authors write that they "turn a blind eye" to the issue of well-defined asymptotic states. Here, the reader would benefit if the authors would cite current attempts to address this question.

d) The authors choose to work with a variant of Kruskal-Szekeres coordinates for the background Schwarzschild spacetime. It remains unclear to the reader whether there is a particular reason to work in these coordinates.

e) Appendix B and C are referred to in the main text in reverse order. Moreover, App. C is very extensive. While I appreciate that the authors move this technical part to an appendix, whenever they refer to results in the main text, it would be helpful if they provide reference to a concrete equation in App. C. For instance, it was difficult for me to identify where Eq. (2.7) originates from precisely.

f) I appreciate that Sec 2) contains a comparison to closely related work. However, I found it somewhat difficult to follow which part concerns which convention. To give a concrete suggestion, the reader would benefit from a summarizing paragraph at the beginning of Sec. 2) that announces where which convention is addressed.

g) "foor" on p.12 should probably read "for"

h) The reader is left wondering about the significance of Sec 4.1 and 4.2.

i) In Sec. 6, the authors state: "we make an intriguing observation that appears to emerge from the second quantised approach of the present article". It remains unclear to me what they refer to as "the second quantised approach".

---

## Round 2 · Author Response

We have addressed all the issues raised by the two referees and list the changes made below. Moreover, we will explain some of the issues raised by the referees as responses to their reports (in such cases where a modification to the submission was not necessary). Since both referees have deemed the article to warrant publication in SciPost Phys. with minor changes (which we believe to have satisfactorily implemented), we request that the paper be accepted for publication.

---

## Round 2 · List of Changes

We have addressed all the issues raised by the two referees. Here, we list the changes made: 1) We have added a clarifying note at the top of page 11 comparing the dynamical mode we have with that in the literature. Limitations to Schwarzschild added in Sec 7. 2) There is no first principle derivation or argument that we are able to propose for why the scattering energy. However, as we explain in Sec 6, since the scattering process conserves momentum, adding a scattering energy of the black hole mass results in an entropy associated with the the typical entropy for decay of a state with energy E = M_BH into a large number of particles, analogous to black hole evaporation. We have added this clarifying comment in the discussion below eq (6.3) to make the importance of the scattering energy E = M_BH clearer. 3) We have included a reference to lectures with a pedagogical overview of the time scales associated with he black hole in the introduction. 4) We added a reference (in the first bullet point in the Shortcomings paragraph in the Introduction) where the problem of asymptotic states in black hole backgrounds is indirectly addressed by resorting to lower-dimensional AdS/CFT. 5) The metric in standard Kruskal-Szekeres coordinates has unwarranted numerical factors on the horizon. The variant we use ensures that the metric on the horizon is such that A(r=R)=1. We have added this comment below (A.2). 6) We reversed the order of appendices. There was a typo in (2.7) which has been fixed, and we referenced the equation in the appendix it originates from, in the text below eqn 2.8. 7) We made a note on the possible ambiguity in choice of conventions at the top of pg. 11. 8) We have added a clarifying sentence below Eq (4.23) about the usefulness of the simplified vertex in some elastic amplitudes (despite its uselessness for most observables). 9) We clarified the distinction we would like to point out between 't Hooft's quantum mechanics approach vs the second quantised approach being taken in the present article in footnote 8.

---

## Editorial Decision

refereeing_in_preparation